



# Verification of ECMWF System4 for seasonal hydrological forecasting in a northern climate

Rachel Bazile[1], Marie-Amélie Boucher[1], Luc Perreault[2], and Robert Leconte[1]

[1]Département de génie civil, Université de Sherbrooke, 2500 Boul. de l'Université, Sherbrooke, Québec, J1R 2R2, Canada
[2]Institut de Recherche d'Hydro-Québec (IREQ), 1800 boul. Lionel-Boulet, Varennes, Québec, J3X 1S1, Canada

*Correspondence to:* rachel.bazile@gadz.org

**Abstract.** Hydro-power production requires optimal dam management. In a northern climate, where spring freshet constitutes the main inflow volume, seasonal forecasts can help to establish a yearly strategy. Long-term hydrological forecasts often rely on past observations of streamflow or meteorological data. Another alternative is to use ensemble meteorological forecasts produced by climate models. In this paper, those produced by the ECMWF (European Center for Medium-Range Forecast)'s

System 4 are examined and bias is characterized. Bias correction, through the linear scaling method, improves the performance of the raw ensemble meteorological forecasts in terms of Continous Ranked Probability Score. Then, three seasonal ensemble hydrological forecasting systems are compared: 1) the climatology of simulated streamflow, 2) the ensemble hydrological forecasts based on climatology (ESP) and 3) the hydrological forecasts based on bias-corrected ensemble meteorological forecasts from System4 (corr-DSP). Simulated streamflows are used as observations. Accounting for initial conditions is valuable

even for long-term forecasts. ESP and corr-DSP both outperform the climatology of simulated streamflow for lead-times from 1-month to 5-month depending on the season and watershed. Corr-DSP appears quite reliable but sometimes suffer from under-dispersion. Integrating information about future meteorological conditions also improves monthly volume forecasts. For the 1-month lead-time, a gain exists for almost all watersheds during winter, summer and fall. However, volume forecasts performance for spring is close to the performance of ESP. For longer lead-times, results are mixed and the CRPS skill score is

close to 0 in most cases. Bias-corrected ensemble meteorological forecasts appear to be an interesting source of information for hydrological forecasting.

## 1 Introduction

Hydro-power production planning typically requires inflow forecasts to reservoirs at different lead-times. Whereas short term forecasts are used for day to day planing, sub-seasonal (1 to 3 months) to seasonal (up to 6 month) forecasts are used to establish

a yearly strategy. Improving the skill of hydrological forecasts at the sub-seasonal to seasonal scale is thus essential. According to DelSole (2004): "A system is said to be unpredictable if the forecast distribution, which gives the most complete description of the future state based on all available knowledge, is identical to the climatological distribution, which describes the state in the absence of time lag information". Hence, the advent of seasonal meteorological forecasts that are more informative than climatology could support water managers in their decision process.





Probabilistic forecasts are necessary to quantify uncertainty about future hydrological conditions. It is especially true for long-term forecasts, as uncertainty grows with lead-time. Operationally, several methods exist to produce sub-seasonal to seasonal hydrological forecasts. They can be broadly divided into two main categories: statistical forecasting and ensemble-based forecasting (Yuan et al., 2015). However, hybrid methods also exist.

Statistical methods can take advantage of relationships between past and future streamflow persistence, (e.g. Svensson, 2016) or between streamflows and teleconnections indices. Examples include Bayesian inference (e.g. Wang et al., 2009) and multiple regression (e.g. Moradkhani and Meier, 2010; Sveinsson et al., 2008).

Ensemble-based forecasting is a widespread uncertainty assessment technique (Cloke and Pappenberger, 2009). Ensemble forecasts comprise different potential future scenarios also called 'members'. One possible method to obtain hydrological en-

10 sembles is to provide ensemble meteorological forecasts as inputs to one or several hydrological model(s). Each meteorological scenario leads to one hydrological scenario (member). From these members, different techniques exist to derive probabilistic forecasts (Bröcker and Smith, 2008).

For long forecasting horizons, the simplest type of ensemble forecasts is the climatology of streamflow, hereafter called 'Historical Streamflow Prediction' (HSP). This naive forecasting method is by definition a reliable forecasting system but of

15 course its resolution can be improved. Even if this kind of forecasting system does not show any predictability, it accounts for different plausible hydrological scenarios based on the past. A simple alternative method, proposed by Day (1985), is called 'Extended Streamflow Prediction' (ESP). To produce ESP, past meteorological observations are considered as equiprobable potential future meteorological scenarios. If the historical record is long enough, climatology provides a reliable estimation of the distribution of future meteorological conditions, including some extreme scenarios. The main advantage of ESP relative to

20 HSP is that it allows accounting for the current hydrological initial conditions. Several studies have shown that state variables such as soil moisture or snow water equivalent can provide relevant information to extend predictability for lead-times from 1 to several months ahead (e.g. Wood and Lettenmaier, 2008; Shukla et al., 2013; Yang et al., 2014; Yuan et al., 2016). The influence of initial conditions depends on the period of the year and on the location of watersheds (e.g. Yossef et al., 2013). ESP are intuitively appealing, since they are coherent with a natural tendency of humans to judge actual situations according

to their memory of past experiences. Moreover, ESP allow practitioners to condition streamflow scenarios only on selected past meteorological scenarios if they wish, for instance by using only the most extremes historical scenarios. Because of their simplicity and efficiency, both ESP and HSP are popular among operational agencies for forecasts from several days to weeks to months (e.g. García-Morales and Dubus, 2007) and still arouse interest as a forecasting system (e.g. Singh, 2016).

However, given the current context of climate change, some past meteorological and hydrological data might not be rep-

30 resentative of plausible future conditions. In Nordic contexts, it is expected that climate change will gradually modify the repartition of rain and snowfall during the year, for instance. For the province of Quebec in Canada, climatic projections anticipate a rise in temperature and precipitations (Ouranos, 2015). It is expected that these changes will modify hydrological conditions both at the annual and intra annual scales. Indeed, higher winter streamflows, earlier spring freshet and longer periods of low streamflow during the summer are expected (Guay et al., 2015). In central Sweden, climate change will also affect

the seasonality of streamflow, mostly by decreasing the mean snow water equivalent and the mean annual runoff (Xu, 2000).



During the past decade, sub-seasonal to seasonal ensemble meteorological forecasts produced by dynamic climate models have undergone constant improvements and it is worth questioning their usefulness for long lead-times inflow forecasting. A dynamic climate model is an atmospheric model, sometimes coupled with an ocean model. Considering the interactions between the atmosphere and oceans allows for modeling long-term phenomenon such as El-Niño and La Niña phases of the

5    ENSO cycle. For instance, according to Kim et al. (2012), for winter in the northern hemisphere, both the European Centre for Medium-Range Weather Forecasts (ECMWF) System4 and the National Centers for Environmental Prediction (NCEP) Climate Forecast System Version 2 (CFSv2) accurately reproduce El Niño/La Niña phases. Temperature variations are more difficult to capture. Regarding the ECMWF System4, Weisheimer and Palmer (2014) assessed the performance of 2m temperature and precipitation forecasts throughout the world. The reliability of forecasts vary from "perfect" to "dangerous", depending on the

10    month of the year, the variables and the location.

Meteorological forecasts from dynamic climate models can be used to produce hydrological forecasts, hereafter called "Dynamical Streamflow Prediction" (DSP). However, according to previous studies, their potential for hydrological purposes is highly variable, depending on the location and the context. Luo and Wood (2008) compared forecasts from the NCEP Climate Forecast System (CFS), multimodel forecasts from a combination of CFS and seven models from the DEMETER database and

ESP for hydrological forecasting on a watershed in Ohio during summer. They found that the multimodel approach is more efficient than a single climate model. Both outperform the ESP approach in terms of Ranked Probability Scores. Mutlimodel approaches improve seasonal forecasts significantly for the 1-month and 2-month lead-times, whereas single model improvements are limited. Across the United States, the ESP approach has been compared to hydrological ensemble forecasts based on NCEP CFSv1 and CFSv2 by Yuan et al. (2013). Their results indicate that CFSv2 improves hydrological forecasting perfor-

mances for the 1-month lead-time, whereas CFSv1-based forecasts are not very efficient. Similarly, He et al. (2016) compared the performance of climatology (ESP) and CFSv2 for a single watershed in the Sierra Nevada. Their results indicate only little improvement when using CFSv2. Some agencies already integrate information from long-term ensemble forecasts into their operational hydrological forecasting systems (e.g. Demargne et al., 2014). One major problem of DSP is that ensemble meteorological forecasts produced by dynamic models suffer from bias. However, in a context of climate change, using dynamical

meteorological forecasts seems intuitively valuable, as they are expected to better represent the current climate, compared to methods based on past climatology (such as ESP). Crochemore et al. (2016) compared different strategies for bias correction of daily precipitation forecasts and evaluated their efficiency for hydrological forecasting over 16 watersheds in France. They show that correcting precipitation forecasts does indeed translate into an improvement of hydrological forecasts. However, they also show that simple bias correction methods, such as linear scaling, are as efficient as more sophisticated methods.

The goal of this study is to evaluate the potential of Dynamical Streamflow Prediction (DSP) in terms of predictability improvement for long term streamflow forecasting, compared to Historical Streamflow Predicition (HSP) and Extended Streamflow Prediction (ESP). More specifically, long-term meteorological and hydrological forecasts are assessed for 10 northern watersheds in the province of Québec in Canada. Those watersheds are all exploited for hydro-power production. Therefore, skillful long lead-time forecasts are crucial for optimal water management, especially for anticipating and exploiting the large

inflow to reservoirs during spring melt.





The paper is organized as follow. After describing the context in section 2, details regarding the case studies and available data will be given in section 3. The forecasts verification methodology is presented in section 4. Results are presented and discussed in section 5 and conclusions are drawn in section 6, which also identifies potential research avenues for future studies.

## 2 Hydro-meteorological context of the study

### 2.1 Watersheds

The ten watersheds used as a testbed in this study are all located in the province of Québec and exploited by Hydro-Québec to generate hydro-power. Together, they represent more than 8750 MW (Hydro-Québec, 2015), as the outlet of each watershed is a hydro-power reservoir. Figure 1 illustrates the geographical location of the ten watersheds. Two of them are located in the southern portion of the province, whereas the others are located in the central portion. In addition, some watersheds are part of larger hydro-power production complexes. For instance, watersheds 4, 7, 8 and 9 compose the Manicouagan complex and watersheds 5, 6 and 10 are part of the La Grande complex.

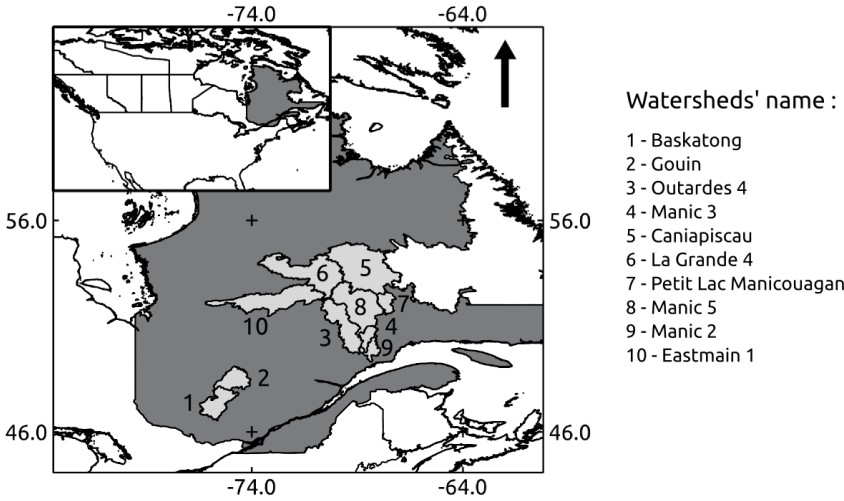

**Figure 1.** Geographical location of the watersheds used in this study

The streamflow regime of the ten watersheds is dominated by a northern climate, which induces snow accumulation and low streamflow during winter (December to February), followed by high streamflow during spring. The exact timing of the spring freshet for a particular watershed is a function on its latitude, its physiographic characteristics, such as slope and orientation, but also of the meteorological conditions that prevail during a particular year. Watersheds located in the southern portion of the province generally produce their highest streamflows in March or April, whereas those located in the central and northern





parts reach their maximum streamflows in May or June. The total volume of runoff associated with the spring freshet obviously depends on the accumulated snow pack during the winter season. Most watersheds also exhibit high streamflow during fall on most years, when evapotranspiration is low and soil is saturated. In fact, whereas temperatures generally show large variability during the winter, the spatio-temporal variability of precipitation is higher during summer and fall. Table 1 presents the hydro-meteorological characteristics of the ten watersheds under study.

**Table 1.** Hydro-meteorological characteristics of the watersheds illustrated on Figure 1

| ID (see Figure 1 | Area $(km^2)$ | Mean min. temp. (°C) | Mean max. temp. (°C) | Mean annual precip. (mm) | Mean min. stream-flow $(m^3/s)$ | Mean max. stream-flow $(m^3/s)$ | Mean date of the max. spring stream-flow | Proportion of the spring freshet volume compared to the annual volume (%) |
|---|---|---|---|---|---|---|---|---|
| 1 | 13 057 | -30 | 23.9 | 1018 | 17 | 1359 | 28/04 | 46 |
| 2 | 9 426 | -31.2 | 23.8 | 971 | 11 | 1282 | 11/05 | 40 |
| 3 | 17 119 | -33.6 | 21.7 | 909 | 80 | 1759 | 20/05 | 44 |
| 4 | 4 245 | -31.1 | 21.6 | 970 | 19 | 536 | 18/05 | 45 |
| 5 | 37 328 | -34.7 | 20.5 | 798 | 151 | 3074 | 30/05 | 38 |
| 6 | 28 443 | -36.6 | 20.9 | 806 | 140 | 1384 | 09/06 | 30 |
| 7 | 4 565 | -33.3 | 20.9 | 902 | 12 | 638 | 25/05 | 45 |
| 8 | 24 608 | -33.6 | 20.8 | 901 | 42 | 3275 | 21/05 | 48 |
| 9 | 4 100 | -28 | 21.5 | 1005 | 6 | 537 | 15/05 | 51 |
| 10 | 26 944 | -35.8 | 21.2 | 838 | 109 | 1981 | 23/05 | 37 |

## 2.2 Current operational streamflow forecasting system

The current operational streamflow forecasting system at Hydro-Québec is divided into three distinct stages. The first stage, for short-term forecasting, is an analog method (e.g. Marty et al., 2012) based on the deterministic meteorological forecast from Environment and Climate Change Canada. The analog-based meteorological ensembles are then fed to a lumped conceptual hydrological model (described below). The definition of "short-term" is not fixed but rather varies with watersheds and events. On average, it varies between five to seven days. The second stage, for seasonal forecasting, relies on Extended Streamflow Prediction (ESP, Day, 1985). Observed precipitation and temperature for previous years are considered as plausible future scenarios. Hence, archived observed meteorological conditions for all previous years (since 1950) form an ensemble. This ensemble is concatenated with short term meteorological forecasts and both are used as inputs for the hydrological model. Lastly,





the third and last stage begins when the influence of initial conditions becomes negligible. Observed streamflow for the same Julian day of each available year in the database are then considered as equiprobable long-term forecasts (Historical Streamflow Predictions, see Introduction). The appropriate moment to shift from ESP to HSP is fixed by the forecaster and varies between watersheds. Note that Hydro-Quebec is currently improving its forecasting system by integrating ensemble weather

forecasts with statistical post-processing for short-term forecasting, and by developing a weather generator for medium-term forecasting. This new system is expected to become operational in 2018.

The available archive of past meteorological observations covers the 1950-2015 period. Data include daily minimum and maximum temperature as well as daily rainfall and snowfall. Those variables are only available at the watershed scale, meaning that observations from individual weather stations were spatially aggregated before being archived. Those weather stations are

10 part of a province-wide cooperative network called RMCQ (in French *Réseau météorologique coopératif du Québec*, Lepage and Bourgeois, 2011). The aim of this cooperative network is to pool together data from private and public collaborators. Unfortunately, the number of stations and the interpolation method have evolved over time. At the time of writing, it was not possible to obtain detailed information regarding those successive changes. However, meteorological data is generally of good quality and there is no missing day. Daily data are collected from 6 UTC to 6 UTC for precipitation and temperature and from

15 5 UTC to 5 UTC for streamflow data.

For the purpose of this study, climatology-based ensemble forecasts were built. All available years were used except one, in rotation. For instance, the climatology based forecasts for year 1980 include all years but 1980. This setup is of course different than the operational framework, where the ensemble size grows year after year and the future is unknown. However, the methodology used here allows to maintain a constant ensemble size (64 members). In addition, this produces ensembles that

are free of any possible trend in the time series of climate data. Indeed, all information about past meteorological conditions are used as inputs to the hydrological model.

In all cases, meteorological series are used as input to HSAMI, a lumped conceptual hydrological model described below (in French, Fortin, 2000). This model is based on a series of three linear reservoirs which supply two hydrographs. Snow accumulation and melt are based on a degree-day approach. HSAMI uses daily minimum and maximum temperature, as well

as rainfall and snowfall to compute the mean streamflow at the outlet of the watershed at a daily time-step. The model has 23 parameters that must be calibrated against previous streamflow observations. The sets of parameters used in this study are provided by Hydro-Québec. Modeling performance varies greatly from one watershed to another. The Nash-Sutcliffe efficiencies (NSE) ranges from 0.30 to 0.86 for the 1981-2015 period. Despite low NSE for some watersheds, it was judged appropriate to use these parameter sets rather than recalibrating the model. First, this variation is attributable mostly of the

quality of hydrological data collected before 2000, on which the calibration of the model is based. Second, since the goal of this study is to assess the influence of meteorological forecasts on hydrological forecasts, simulated streamflows are used as pseudo "observations" in the verification process, and therefore a perfectly well calibrated model is not required (also see Section 4 for details).

The next section describes the ECMWF System4 that is explored in this study as a potential replacement for the current

operational forecasting system.



## 3 An alternative system for seasonal forecasting based on long-term dynamical climate modeling : The ECMWF System4

Our hypothesis is that exploiting dynamical meteorological forecasts in the streamflow forecasting chain would improve the latter compared to ESP. The rationale behind this hypothesis is that dynamical meteorological forecasts should be driven

by the current state of the atmosphere at their initialization. Eventually, considering the context of an evolving climate, this could also help hydro-power producers to adapt reservoir and dam management to new situations. The long-range ensemble meteorological forecasts used in this study are produced by the ECMWF (European Center for Medium-Range Forecast)'s System4.

System4 (Molteni et al., 2011) is a global coupled ocean-atmosphere model that officially became operational in 2011.

It is used to produce reforecasts and real-time forecasts, that are both archived. The atmospheric model component, namely the ECMWF IFS (Integrated Forecast System) model (version 36r4) includes a lake model and also involves ozone, volcanic aerosol and solar cycle action. Sea ice is depicted by initial sea ice condition for short lead-times and by observed conditions for the five previous years. The initialization of the atmospheric model is performed using ERA-Interim for the reforecasts and the operational procedure of the ECMWF for real-time forecasts. The ocean model is initialized by the Nemovar ocean

analysis. Ensemble forecasts are produced by perturbing initial conditions. In the current model setup, five members originate from perturbations of ocean wind surface initial conditions, whereas other members originate from sea surface temperature perturbations and stochastic physics. More details can be found on the ECMWF website (ECMWF, 2017).

Real-time forecasts are issued on the first day of each month for the next 215 days (approx. seven months). They are archived and available from 2012 to 2015. A set of reforecasts is also available, from 1981 to 2011. Reforecasts for the months

of February, May, August and November as well as real-time forecasts comprise 51 members. Reforecasts for the other months comprise 15 members. Both archived past real-time forecasts and reforecasts are used in the present study. This allows the extension of the verification data base length, but poses certain challenges in terms of performance assessment, since the number of members vary. In the following, the term "forecasts" will refer indifferently to real-time forecasts and reforecasts.

In the context of this study, the meteorological variables of interest are those that are inputs to HSAMI, namely daily

minimum and maximum temperature as well as total daily precipitation. The original output grid of System4 has a 0.7 degree horizontal resolution for the atmospheric model and around 1 degree for the ocean model at mid-latitudes. Those original resolutions are both too coarse for hydrological applications, as only very few grid points fall inside the watersheds delineations. The original grid was thus downscaled to a 0.1 degree grid through linear interpolation in order to obtain multiple grid points for each watershed. Then, since HSAMI is a lumped model, grid points were averaged to aggregate the information at the

watershed scale. Total precipitation was separated into rainfall and snowfall according to air temperature.

## 4 Forecast quality assessment

As the main goal of long-term hydro-meteorological forecasts is to provide information for seasonal to yearly dam management, decision-makers are generally interested in inflow volumes to reservoirs. Consequently, monthly aggregated variables





are considered. Monthly averages are computed for minimum and maximum temperature. For precipitation and streamflow, monthly cumulative values are considered using calendar months. Many other types of information derived from streamflow forecasts are useful for dam management. Anticipating runoff volume for spring freshet is crucial, as it allows for planning the lowering of the reservoirs to avoid risks of spillage and flooding. The inflow volume for the spring freshet is calculated

between $1^{st}$ March and May 31 for watersheds 1 and 2 and $1^{st}$ April and June 30 for all other watersheds.

Both forecasts and reforecasts are pooled together to assess forecasts performance. Overall, 420 ensemble forecasts are available for verification purposes, as one ensemble forecast is issued on the $1^{st}$ of each month between 1981 and 2015. On the one hand, the verification set should be homogeneous. However, in reality, forecasts characteristics change depending on the period of the year and contradictory behaviors can balance each other out. On the other hand, the verification set should be

as large as possible, in order to ensure statistical significance of the results. Considering these two requirements, skill scores of monthly variables are calculated over seasons. Four seasons are used, namely January-February-March (JFM), April-May-June (AMJ), July-August-September (JAS) and October-November-December (OND). For one season and one lead-time, each set of verification comprises 105 monthly ensemble forecast-observation pairs.

Different numerical scores and graphical tools are used to assess the quality of the aforementioned quantities. The joint

use of several tools is essential for different reasons. First, ensemble and probabilistic forecasts can be evaluated in terms of different attributes and no single score can simultaneously assess them all. Second, examining different attributes can help to pinpoint strengths and weaknesses of competing forecasting systems. According to Gneiting and Raftery (2007), a good probabilistic forecasting system should be reliable and sharp. Reliability refers to the statistical consistency between the predictive distribution and the observation, while sharpness refers to the concentration of the predictive distribution. A reliable

probabilistic forecasting system produces predictive distribution which are unbiased and representative of the true uncertainty underlying the process.These two attributes are important in an operational context, as scenarios are used for decision making.

Forecasts reliability is assessed using the reliability diagram. Confidence intervals computed from reliable forecasts should be in agreement with their definition: the 95% confidence interval, for instance, must include on average 95 observations out of 100. For each nominal confidence level probability from 0.1 to 0.9, the effective frequency of the observation occurrence in the

given nominal interval is calculated. Then, the effective frequencies are plotted against the nominal confidence level probability. Moreover, the probability integral transform (PIT) histogram, which has the same interpretation as the rank histogram described in Hamill (2001) is also used to detect bias and dispersion issues in forecasts. PIT histograms are preferred over rank histogram herein because of the changing number of members (see section 3).

Scoring rules address reliability and sharpness simultaneously. One of the most well-known probabilistic scoring rules,

the Continuous Ranked Probability Score (CRPS, Matheson and Winkler, 1976) is used to assess the overall accuracy of competing forecasting systems. The mathematical expression of the CRPS is given by equation 1.

$$CRPS(p(x), y) = \int (p(x) - H(x < y))^2 dx \qquad (1)$$





where $p(x)$ represents the cumulative predictive distribution of the forecast and $y$ is the observation. $H$ is the step function, which equals 0 when $x < y$ and 1 when $x > y$.

The CRPS skill score (CRPSS) expression is presented in equation 2.

$$CRPSS = 1 - \frac{CRPS_{for}}{CRPS_{ref}} \tag{2}$$

where $CRPS_{for}$ is the mean CRPS of the forecasting system and $CRPS_{ref}$ is the mean CRPS of the reference system (benchmark).

In order to evaluate the potential of ensemble meteorological forecasts, simulated streamflows were used instead of observations in the verification process. Proceeding in this way eliminates concerns about model and parametrization errors, which vary with watersheds and periods of the year. Moreover, after a spin-up period, the initial conditions are not necessarily es-
timated correctly by the hydrological model. Operationally, this is corrected by the forecaster, manually or by an automated data assimilation procedure, so that the simulation matches the observations closely. Since data assimilation falls outside the scope of the present study, using simulated streamflow as a benchmark eliminates this concern. Furthermore, as it is frequently the case for hydro-power reservoirs, observations are not really obtained from gauging stations but rather estimated by a water balance equation applied on each reservoir. The quality of observations also varies across watersheds. Simulated streamflow
series form a complete dataset with no missing data but are subject to errors attributable in a large portion to the hydrological model itself. Consequently, the results presented in section 5 should be interpreted as the potential skill (not the operational one) of meteorological forcings, as if the hydrological model was able to reproduce the watershed's behavior perfectly.

In the following, hydrological forecasts based on bias-corrected forecasts from System4 will be referred to as corr-DSP. Hydrological forecasts based on climatology will be referred to as "Extended Streamflow Predictions", or ESP, and hydrological
forecasts based on the simulated streamflow climatology as simulated 'Historical Streamflow Prediction' or 'sim-HSP'. Moreover, the lead-time refers to the time lag between the emission date of the forecast and the time at which the forecast is valid. For instance, a skill score for the January-February-March season for the 5-month lead-time correspond to the performance of the forecasts issued 5 months earlier, in August, September and October.

## 5   Results

### 5.1   Bias characterization and correction

Raw forecasts from ECMWF System4 suffer from biases. Bias is calculated as shown in equation 3 for temperature and equation 4 for precipitation.

$$Bias_{Add} = \frac{1}{N} \sum_{k=1}^{N} (\overline{x} - y) \tag{3}$$





$$Bias_{mult} = \frac{\frac{1}{N}\sum_{k=1}^{N}\overline{x}}{\frac{1}{N}\sum_{k=1}^{N}y} \tag{4}$$

where $\overline{x}$ is the mean of the ensemble forecast, $y$ the observation and $N$ the number of forecast-observation pairs in the verification set. Bias is additive for monthly mean minimal and maximal temperature: it is the mean error. For monthly precipitation, bias is the ratio of the forecasts mean to the mean observed accumulation.

5    Figure 2 shows the bias for monthly forecasts by season, lead-time and watersheds.

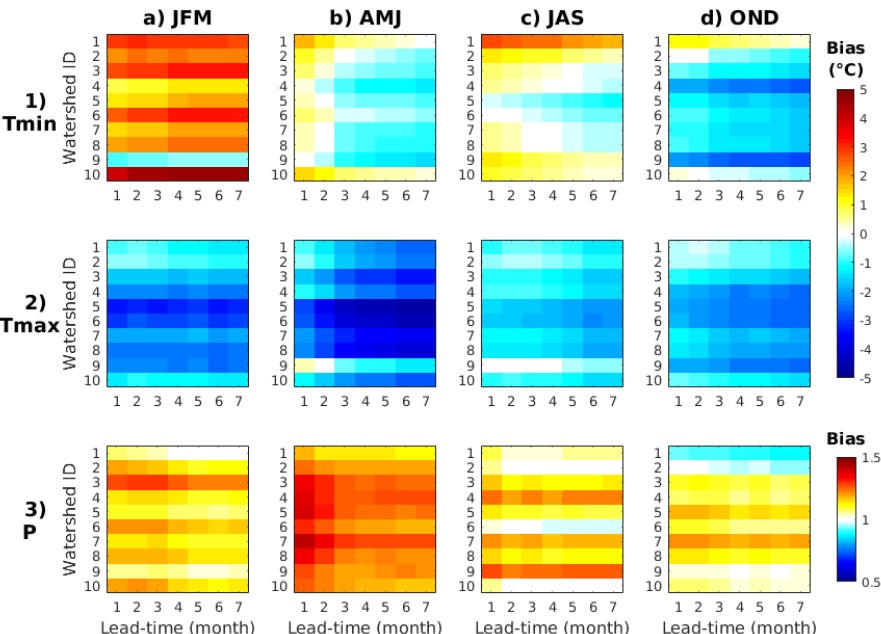

**Figure 2.** Bias for forecasts of 1) monthly mean minimal temperature, 2) monthly mean maximal temperature and 3) monthly accumulated precipitation for the 10 watersheds under study as a function of seasons and lead-times.

Monthly mean maximal temperature forecasts exhibit a cold bias for all watersheds, seasons and lead-times. This cold bias increases with the lead-time. Bias for the mean minimal temperatures changes depending on the season. For almost all watersheds, raw forecasts display a warm bias during winter and a cold bias during fall. During spring and summer, temperature forecasts are almost unbiased. Monthly accumulated precipitation are overestimated most of the time, especially during spring.

10    As shown in Figure 2, raw forecasts clearly need to be bias corrected. As mentioned above, Crochemore et al. (2016) have shown that the simple linear scaling method provides results comparable to the more complex distribution mapping. Hence, in this study, daily precipitation forecasts are corrected by linear scaling based on monthly bias. Bias is estimated separately for each lead-time and month of the year. A leave-one-year-out procedure is used, which consists in excluding the forecast to correct from the bias evaluation process. For a given forecast, all other forecasts issued on the same day of the year are used



to quantify the bias, calculating the mean of the errors between the ensemble mean and the observation. Multiplicative bias for precipitation and additive bias for temperature are calculated for each monthly forecast. The computed bias is finally used to correct the original daily forecasts.

Figure 3 presents the CRPS skill score (CRPSS) of bias corrected forecasts. with The raw ensemble forecasts is taken as the
5   reference (see equation 2). A CRPSS above 0 (from yellow to red) indicates that bias correction improves the original forecasts, whereas a CRPSS below 0 (from light blue to dark blue) indicates a deterioration of the forecasts.

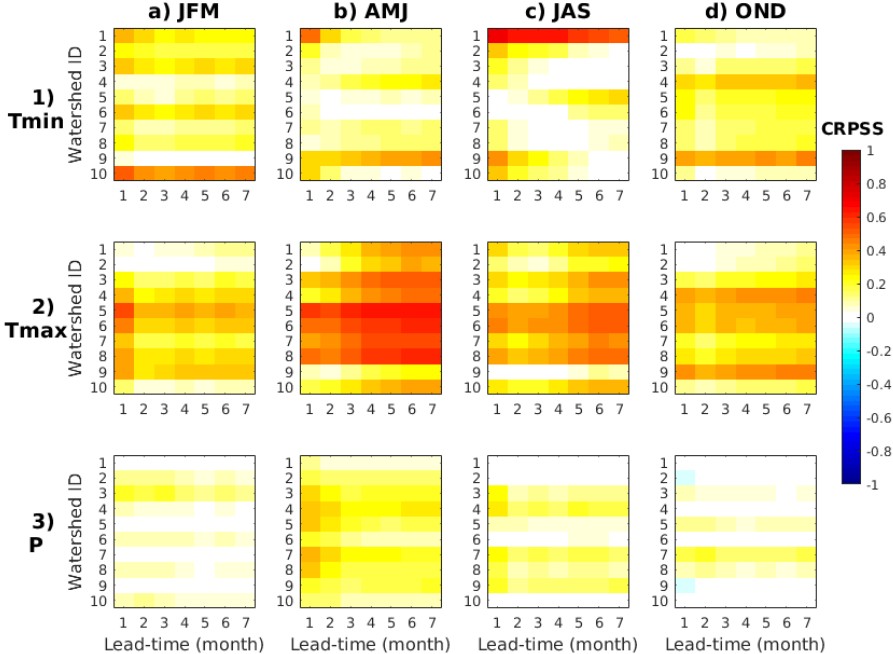

**Figure 3.** CRPSS of bias corrected System4 forecasts compared to raw ensemble forecasts of 1) monthly mean minimum temperature, 2) monthly mean maximum temperature, 3) monthly cumulative precipitation by watersheds, seasons and lead-times.

Bias correction is found effective, as it does improve meteorological forecasts according to the CRPSS. The effect of linear scaling for monthly volume forecasts is not homogeneous throughout the year and it also varies among watersheds. Bias correction is particularly efficient for periods with substantial bias, such as monthly aggregated precipitation during spring.
10   Bias correction of monthly maximum temperature is also efficient for all months, watersheds and lead-times. However, for precipitation during winter and fall, bias correction does not improve the CRPS noticeably. This is likely because bias during those months are generally small.




## 5.2 Performance of ensemble forecasts

### 5.2.1 Bias-corrected meteorological ensemble forecasts against climatology

Figure 4 presents the CRPSS of bias-corrected ensemble forecasts with climatology taken as the reference.

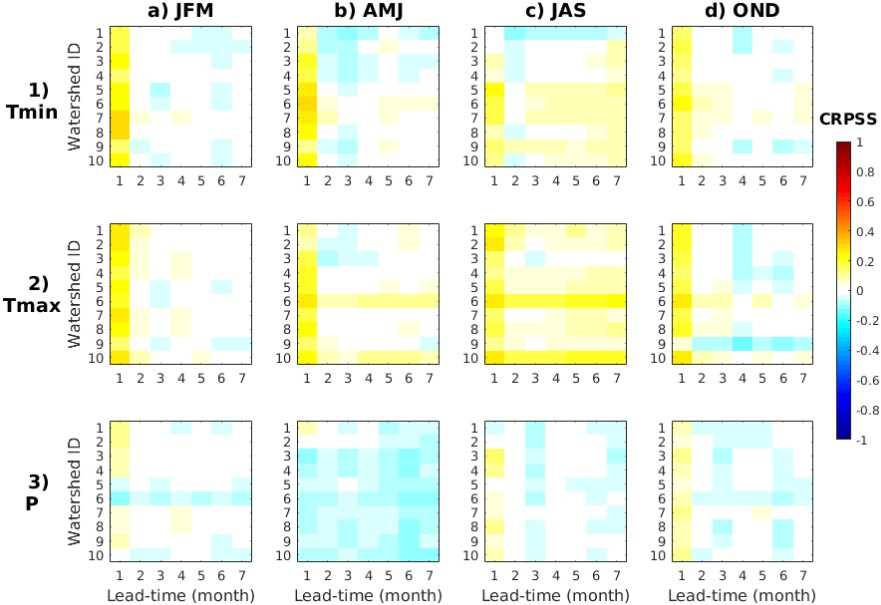

**Figure 4.** CRPSS of bias corrected System4 forecasts compared to climatology for 1) monthly mean minimum temperature, 2) monthly mean maximum temperature, 3) monthly cumulative precipitation by watersheds, seasons and lead-times.

For almost all watersheds and seasons, both mean minimum and maximum temperatures outperform climatology for the 1-
5  month lead-time. However, for longer lead-times, only temperature forecasts during summer can provide a little improvement over climatology.

Precipitation is known to be less predictable than temperature and the CRPSS confirms this insight. For the 1-month lead-time, CRPSS results are mixed. Ensemble forecasts do have some skill for certain watersheds during winter, summer and fall (for instance watersheds number 3 and 4), whereas CRPSS indicate that climatology is more skillfull during spring. For longer
10  lead-times, according to the CRPSS, climatology always outperforms forecasts.

Hydrological conditions depend mostly on precipitation. However, in a northern environment, temperatures are also impor-
tant, especially during winter and spring. In fact, during these periods, temperature defines the type of precipitation (snow or rain). It also drives snow pack maturation and the characteristics of the spring freshet (early/late and fast/slow). During summer, temperature controls evapotranspiration. In these conditions, it is valuable to assess the performance of hydrological forecasts
15  produced by both systems: corr-DSP (based on bias-corrected System4 forecasts) and ESP (based on climatology).





### 5.2.2 Monthly inflow volume forecasts

Figure 5 shows the performance of ensemble forecasts for inflow volumes compared to sim-HSP (simulated streamflow clima-
tology) and ESP. Their performance is assessed using the CRPSS.

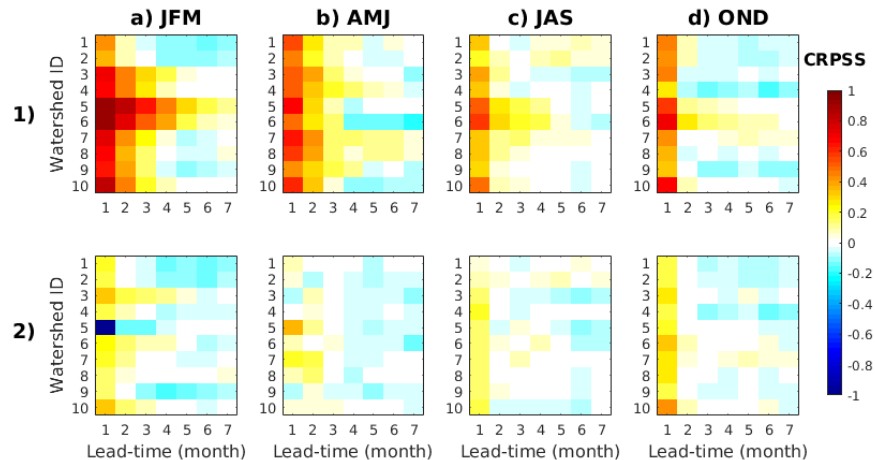

**Figure 5.** CRPSS of ensemble forecasts of monthly inflow volume to reservoirs, produced by corr-DSP compared to 1) sim-HSP and 2) ESP.
CRPSS are shown by watersheds, seasons and lead-times.

The first row of Figure 5 shows the improvement of corr-DSP over sim-HSP. Those results reflect the gain in performance

that could be achieved by considering initial conditions as well as information about bias-corrected meteorological forecasts
from System4. For all seasons and watersheds, it is valuable to use meteorological ensemble forecasts to produce monthly
inflow volume forecasts instead of using simulated climatology (sim-HSP). The critical lead-time, namely the lead-time beyond
which sim-HSP performs better than corr-DSP, depends on the period of the year. More specifically, inflow volume forecasts
for summer and fall do not show much skill beyond the 1-month lead-time. However, forecasts for winter and spring can be

predicted fairly accurately several months ahead when using System4 rather than sim-HSP.

When comparing corr-DSP with ESP (second row of Figure 5), the CRPSS reflects the advantage of integrating meteorolog-
ical information from ensemble forecasts into the hydrological model. The benefit is clear for the 1-month lead-time, except
during spring.The reasons why the CRPSS is not as good during spring as for other seasons could include a change in the
influence of initial conditions during the different seasons and the lack of skill of precipitation forecasts. Indeed, streamflow is

more variable during spring than during the rest of the year.

In general, corr-DSP outperforms ESP for the 1-month lead-time for watersheds number 5 and 7. Predicting monthly volume
during summer and fall more than one month in advance is difficult and both ESP and corr-DSP exhibit comparable skill.
Finally, for some watersheds during the winter months, corr-DSP improves the predictability of monthly volumes compared to
ESP. Watersheds 3, 5 and 7 reflect different CRPSS behaviors, especially for winter and spring months. Thus, for the remaining

of the analysis, special attention is given to those three specific watersheds.





Figure 6 presents the reliability diagrams of corr-DSP for three specific watersheds (number 3, 5 and 7, see Figure 1) by seasons and lead-times. The same diagrams were plotted for all ten watersheds, but results are shown only for those three because, as mentioned above, they reflect specific behaviours worthy of investigation. Furthermore, in retrospect it was found that reliability diagrams for the other watersheds displayed characteristics quite similar to those that are presented. Reliability
5   plots were also obtained for Sim-HSP and ESP but they are, by definition, reliable. Hence, those plots are not shown.

As can be seen on Figure 6, the reliability of corr-DSP monthly inflow volume forecasts changes over seasons. The forecasts produced for the fall season are the most reliable, as the effective probabilities computed from forecasts are close to nominal probabilities for all lead-times. Forecasts do not display strong dispersion issues. However, small differences between the effective probability and the nominal probability remain. Indeed, even if a bias correction was applied to raw meteorological
10  forecasts, some biases can still remain and propagate to hydrological forecasts. In addition, bias correction affects the dispersion of precipitation forecasts, and in turn the dispersion of hydrological forecasts. Figure 7 shows the PIT histograms for the three watersheds for the 1-month lead-time. As mentioned above, some bias are visible in some cases, such as for watershed number 3 during the winter (1a) or underdispersive behavior during the spring (1b).

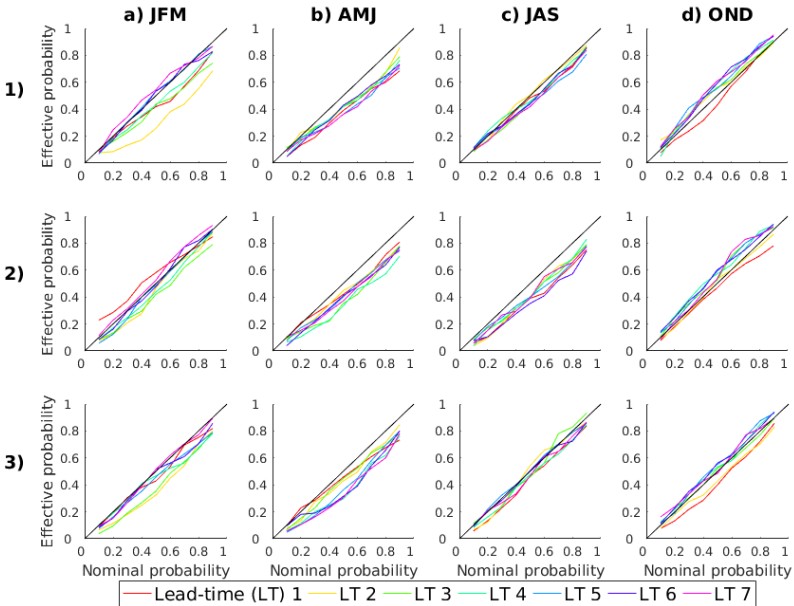

**Figure 6.** PIT histograms for 1) watershed number 3, 2) watershed number 5 and 3) watershed number 7.





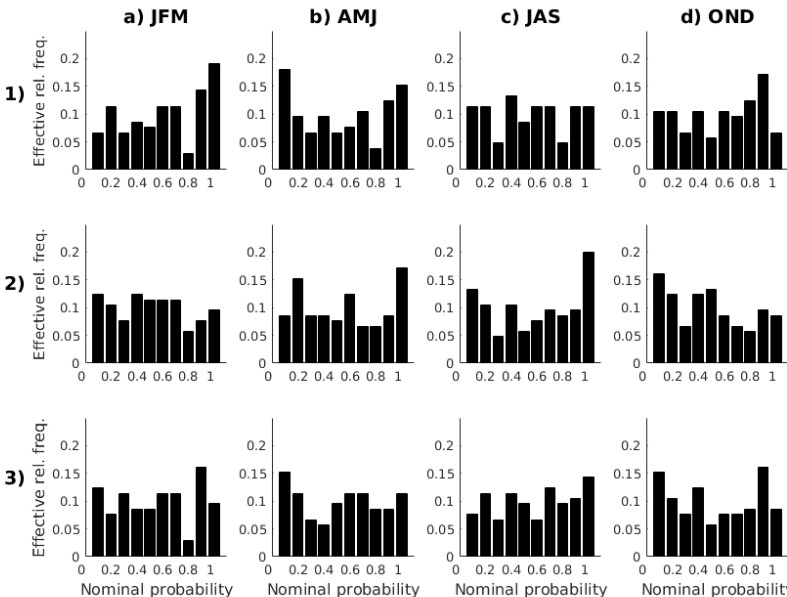

**Figure 7.** Rank histograms for 1) watershed number 3, 2) watershed number 5 and 3) watershed number 7.

### 5.2.3 The case of inflow volume forecasting during spring freshet: an example for watersheds number 3, 5 and 7

Anticipating the inflow volume to the reservoirs is crucial for winter dam management. During the winter, reservoirs are partly emptied to ensure storage space for the inflow volume that is expected during the spring freshet. Consequently, good forecasts of inflow volumes are valuable. The spring freshet volume is computed by cumulating daily forecasted volumes over three

5    months periods. The specific time period associated to the spring freshet varies from one watershed to another, mainly because of geographical location. Watersheds have been clustered into two groups. Watersheds 1 and 2, located in the south, have earlier spring freshet. For those two watersheds, the spring freshet period is defined from March 1st to the end of May. For the other watersheds, the spring freshet occurs between the April 1st and the end of June.

Figures 8, 9 and 10 present forecasts for the spring freshet volume from (a) ESP and (b) and corr-DSP for three watersheds

10   (number 3, 5 and 7). For ESP and corr-DSP, the dispersion of forecasts increase with the lead-time. For those three watersheds, ESP exhibit a larger dispersion at all lead-times. Extreme meteorological scenarios from past years lead to possible extreme hydrological scenarios.In some cases, such as the spring of 1993 for watershed number 3 at lead-time 1, corr-DSP provided very poor forecasts that missed the spring freshet almost entirely, whereas the ESP is much more successful (the observed volume is included in the boxplot). This issue could be explained from a bias-correction problem. Indeed, even if bias-correction was

15   applied, some biases in corr-DSP might remains and further propagate to hydrological forecasts. It is also possible that the bias-correction method performs better for some years, lead-times and watersheds than others. Similar figures were obtained





for the other watersheds and the general conclusions for those figures are that corr-DSP exhibit a lower dispersion than ESP, leading observations to fall outside the boundary of the predictive distribution too often.

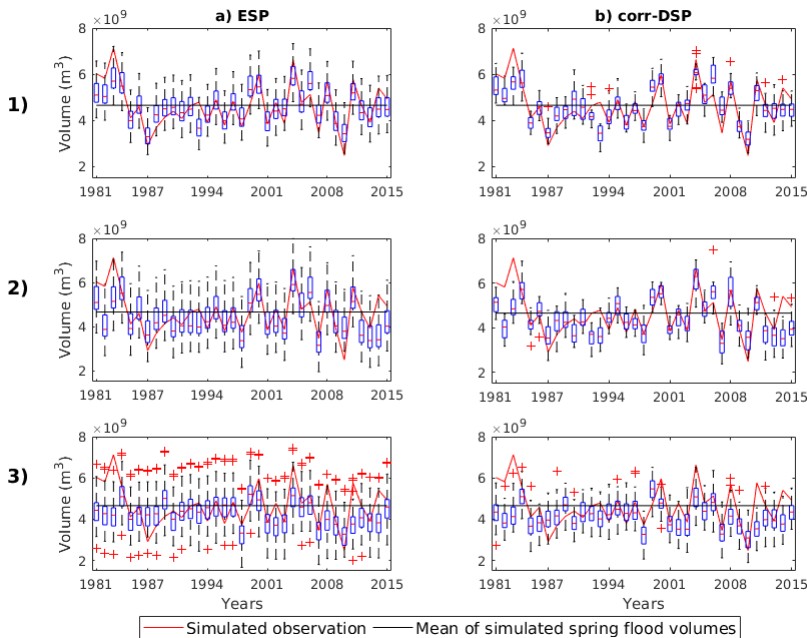

**Figure 8.** Spring freshet volume forecasts at 1) 1-month lead-time, 2) 2-month lead-time and 3) 3-month lead-time for watershed number 3. The boxplots represent the ensemble forecasts for one given spring freshet and the red line the corresponding simulated observation.

Figure 11 presents the boxplot of the CRPS for the 35 spring freshet events between 1981 and 2015, for all watersheds and the three forecasting systems: sim-HSP, ESP and corr-DSP. In addition to the forecasts issued the first day of the spring freshet period (lead-time 0-month), three lead-times are considered: 3, 2, and 1 month(s) before the first day of the spring freshet. In most cases, the CRPS for corr-DSP displays a higher variability than the CRPS for ESP. Depending of the year, corr-DSP can have a better or worse performance than ESP and even sim-HSP. Hence, resorting to sim-HSP, the simplest "forecasting" system of all three (arguably a forecasting system at all) can be advantageous for some watersheds for which the predictibility is low. For instance, for watershed 6 at lead-times 2 and 3 months, sim-HSP outperform all other techniques. However, for lead-times 1 and 0, ESP and corr-DSP have lower CRPS in the majority of cases. Very high (poor) CRPS can result from both dispersion and bias issues.





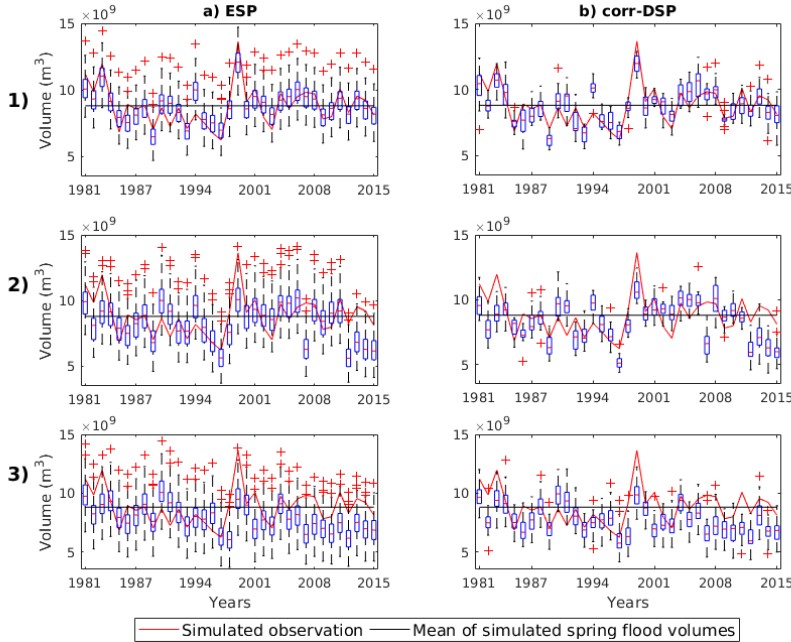

**Figure 9.** Spring freshet volume forecasts at 1) 1-month lead-time, 2) 2-month lead-time and 3) 3-month lead-time for watershed number 5. The boxplots represent the ensemble forecasts for one given spring freshet and the red line the corresponding simulated observation.

## 6 Conclusion

The objective of this study was to compare the performance and the behavior of three hydrological forecasting systems for 10 watersheds in a northern climate (Québec, Canada). The three forecasting systems consist of HSP (streamflow climatology), ESP (forecasts based on meteorological climatology) and DSP (forecasts based on ensemble meteorological forecasts from

ECMWF System4). Streamflow simulations were used to build the streamflow climatology of each watershed. Simulated streamflows were also used as pseudo observations in order to avoid considering hydrological model errors in the analyse.

In the context of this study, it was found that ensemble meteorological forecasts from System4 suffer from biases (see Figure 2). However, a bias correction, performed using the linear scaling method, results in an improved performance of ensemble meteorological forecasts, as assessed by the CRPS (see Section 4). Monthly mean minimal and maximal temperature forecasts

outperform climatology for the 1-month lead-time. The predictability extends to several months in specific cases (watersheds and seasons). Monthly accumulated precipitation are less predictable. In fact, ensemble meteorological forecasts do not have significant skill when it comes to forecasting monthly precipitation during spring. For other seasons, they slightly outperform climatology for the 1-month lead-time.

Still, according to the CRPS, bias-corrected ensemble meteorological forecasts were found to be a useful source of informa-

tion to improve monthly volume forecasts (see Figure 5), especially for the 1-month lead-time. This is likely due to temperature





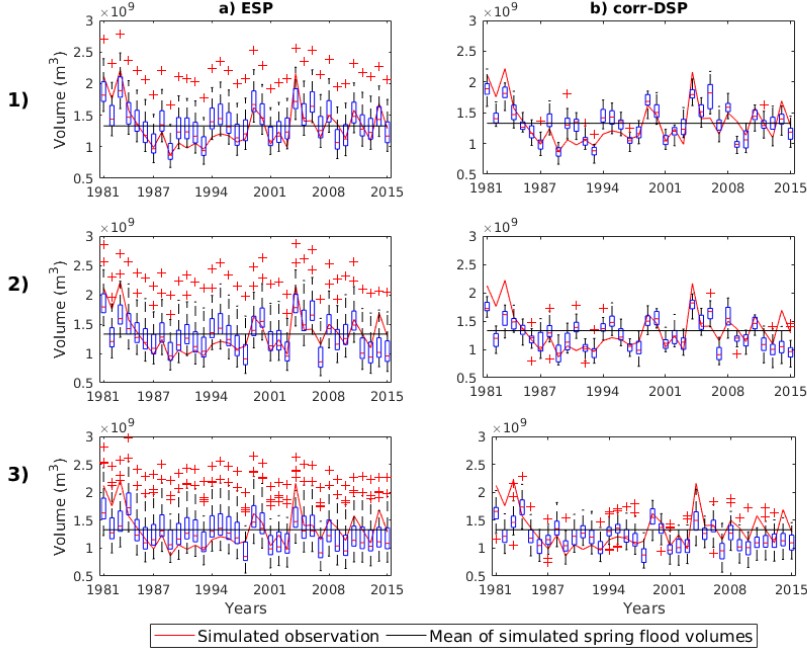

**Figure 10.** Spring freshet volume forecasts at 1) 1-month lead-time, 2) 2-month lead-time and 3) 3-month lead-time for watershed number 7.The boxplots represent the ensemble forecasts for one given spring freshet and the red line the corresponding simulated observation.

forecasts more than precipitation forecasts, as mentioned above. Regarding the particular case of forecasts for summer and fall, the CRPS of corr-DSP outperforms the CRPS of both sim-HSP and ESP for the 1-month lead-time. Predictability of monthly volume for winter and spring months extends up to 3 months against simulated climatology. The CRPSSS between corr-DSP and ESP is lower than the CRPSS between corr-DSP and sim-HSP. However, corr-DSP show some skill from 1-month lead-

time up to 2 or 3-month lead-times for some watersheds (number 3, 7 and 10) during the winter. Monthly forecasts based on System4 are less reliable than ESP, and this possibly originate from bias propagation or dispersion issues. Results for spring are mixed: the forecasting performance during spring freshet varies from one watershed to another. In general, the CRPS of corr-DSP is more variable than the CRPS of ESP.

    Furthermore, skill scores are subject to sample uncertainty. In this study, each skill score computation is based on almost

one hundred forecasts. The size of the verification set thus remains a limit for assessing the significance of the verification results. For the 1-month lead-time, the quality of corr-DSP is clearer and for longer lead-times, the CRPSS of corr-DSP compared to ESP tends to 0. Consequently, in this study and for long lead-times, it is not clear that bias-corrected seasonal meteorological ensemble forecasts from a dynamical model (System4) can completely replace ESP. However, they provide substantial complementary information to produce long-term hydrological forecasts, as shown by the special case of forecasting

the inflow volume associated to the spring freshet (see Section 5.2.3). An analysis on the economic value of forecasts for hydro-power production, for instance using a reservoir operation model based on stochastic dynamic programming, would be the next



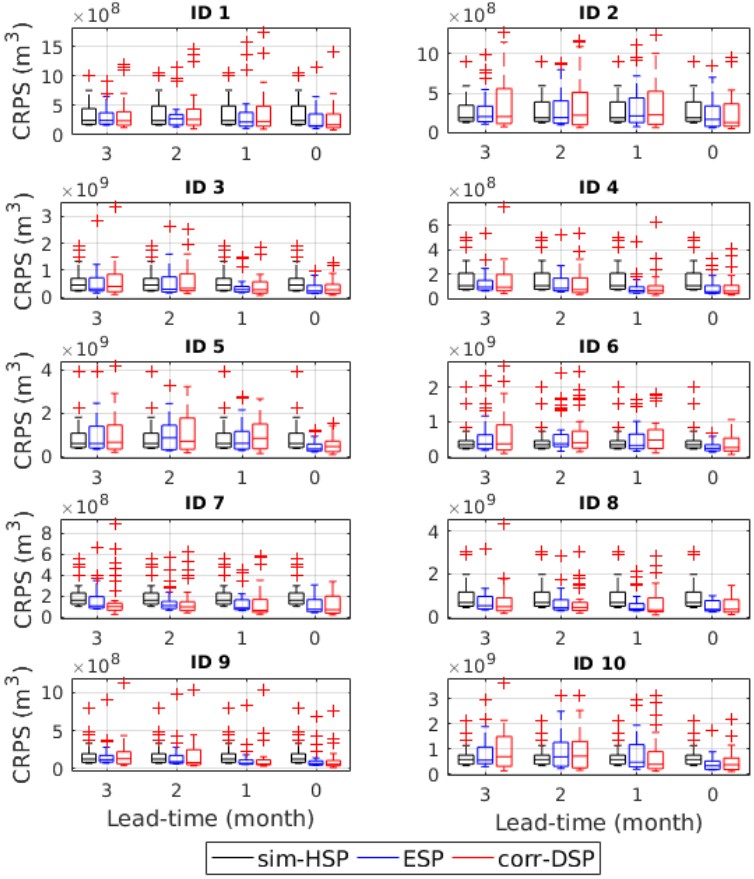

**Figure 11.** Boxplot of the CRPS for the 35 spring freshet events between 1981 and 2015 for all watersheds and the three forecasting systems: sim-HSP (black), ESP (blue) and corr-DSP (red).

logical step. It would allow to determine whether or not the differences observed here between the three concurrent forecasting systems are indeed significant for water management.

Finally, ECMWF's System4 does not include any sea-ice model (ECMWF, 2017). This could limit to improve weather predictability in mid-latitudes. Other providers of ensemble meteorological forecasts exist and a multi-model approach could

5    improve the skill of the seasonal forecasts. Moreover, according to our results, the 1-month lead-time is were the most significant gain could be achieved by dynamical models over climatology (ESP). For that purpose, the newly available Sub-seasonal to Seasonal (S2S) database (Vitart et al., 2017), that gathers ensemble forecasts from different agencies at the sub-seasonal scale (from 1 up to 60 days) could be explored. In fact, the forecasts available in S2S are especially tailored for the 1 to 2-months lead time, and hence could have superior skill for hydrological applications than System4 which, as shown here, can

10    often lead to better 1-month ahead streamflow and volume forecasts than ESP. This new database would also enable future



studies to explore multi-model forecasting approaches at long lead-times and assess the ability of such approach to extend the limits of predictability. Finally, as proposed by Yuan et al. (2014), forecasts for different lead-time would need to be efficiently joined together in a seamless way, and there is also much to explore in this regard.

## 7 Data availability

Unfortunately, the data used in this study is not publicly available. Data from the ECMWF System4 forecasts are produced by the ECMWF but are not included in the repositories of public datasets. They were provided fo the purpose of this study by Dr. Florian Pappenberger (Florian.Pappenberger@ecmwf.int). Meteorological and streamflow data as well as the watersheds delineation file and the hydrological model are the property of Hydro Québec.

*Author contributions.* Rachel Bazile performed all the computation and prepared all the figures. She wrote the most of the manuscript.
Marie-Amélie Boucher guided the work, providing opinions about the presentation of results and analysis to be done. She also provided codes that were used in the computations. She helped writing the manuscript and revised all versions. Luc Perreault provided codes and further guidance for analyzing the results and preparing the manuscript. Robert Leconte initiated the work, reviewed the manuscript and participated in analyzing the results.

*Competing interests.* The authors declare that they have no conflict of interest.

*Acknowledgements.* This work was funded by a NSERC Cooperative Research and Development grant to Robert Leconte. Rachel Bazile gratefully acknowledges a scholarship from the Fonds de Recherche du Québec Nature et Technologies. The authors wish to thank Florian Pappenberger for providing the forecasts from ECMWF System4 as well as Catherine Guay for her support with HSAMI. The team of operational forecasters at Hydro-Québec, are warmly acknowledged for their insights, especially Fabian-Tito Arandia Martinez, Éric Crobeddu and Marie-Claude Simard.





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
