# Peer review of "Verification of ECMWF System4 for seasonal hydrological forecasting in a northern climate"

_Hydrology and Earth System Sciences, 2017_

## Referee Comment (RC1) · K. Förster (Referee) · 21 Jul 2017

**General comments**

In this research article, the authors present a detailed study on the predictive skill of hydrological ensemble forecasts in 10 watersheds in Québec, Canada. Different methods are employed which are subject to different degrees of complexity. Among these methods, a simple application of historical streamflow data is seen as benchmark for more complex approaches. The second method, the ESP approach is based on historical meteorological data and accounts for initial conditions in each forecast. The initialization using known system states is also relevant in the third approach, a dynamical seasonal forecast method, in which meteorological forcing is obtained from

bias-corrected climate model forecasts (ECMWF's System4). Given a lead time of 1 month, the dynamical approach provides improved skill in terms of Continuous Ranked Probability Score (CRPS), while for longer lead times the predictive skill is similar to the corresponding ensemble forecasts using ESP. For the period of snowmelt in spring, the CRPS is lowest (best) in the case of ESP and the dynamical approach. In some watersheds, however, the first method which provides forecasts using historical streamflow data performs best. This comparison highlights the fact that the predictability is low in some watersheds. The study is interesting, the results are promising and the paper fits very well into both the special issue on "Sub-seasonal to seasonal hydrological forecasting" in particular and HESS in general. The methodology is comprehensively presented and the results are discussed in a balanced way. Related work and relevant references are mentioned and acknowledged. Especially the assessment of added value provided by each increased level of complexity (using streamflow data only -> ESP -> dynamical forecasts) is very useful. Another important point is that the paper presents a specific case study in which operational forecasts have already been issued and new methods are going to be implemented. This might be relevant for other forecasting centers. However, in my opinion, the paper needs a few minor revisions and technical corrections. It's my impression that the section on reliability seems to be detached to a certain degree given that the findings from this analysis are not really considered in the summary. Moreover, the paper would benefit from some additional explanations that might improve comprehensibility. Please find my suggestions below.

**Specific comments**

Page 1, line 9: the abbreviation "corr-DSP" is not explained in this context and might be omitted here

Page 1, line 9: Would it make sense to point out that "Simulated streamflow computed using observed meteorological data is used as benchmark."?

Page 2, lines 12-28: In this section, historical streamflow prediction (HSP) and extended ensemble streamflow prediction (ESP) are presented. In my opinion, some additional explanations might be helpful in this context. You could explain that using HSP is in general possible without using a hydrological model, even though, in particular, you involve the output of a model in your specific case study. ESP, in contrast, does require a hydrological model in order to improve forecasts through explicitly incorporating initial states in the forecasts. The relevance of using hydrological models, as already pointed out, might be helpful in the process of understanding the different methods you apply.

Page 3, line 2: I am not sure whether "questioning" is the appropriate verb in this context. As far as I know it would make sense if you have reason to doubt the usefulness. Instead, using "assessing" might be a better option.

Page 3, line 14: Please add appropriate references of the DEMETER project and also explain the project's acronym.

Page 5, Table 1: Please add mean temperature and mean streamflow if easily available. As mean precipitation is indicated, averages of temperature and streamflow might gain insight into the climate characteristics.

Page 5, line 13-14: Does this mean that short-term forecasts are extended by the ensembles generated using ESP? Please consider rephrasing.

Page 6, lines 27-28: Do the Nash-Sutcliffe values are computed using daily time series?

Page 7, line 18: Forecasts are also computed using one day time steps?

Page 8, line 6: Please indicate why the number of forecasts amounts to 420. 35 years x 3 months x 4 seasons?

Page 8, line 23: Is the term "confidence interval" really correct in this context? As far as I understand, say we consider a reliable forecast of a specific event, a probability of 95% should at best also refer to 95 out of 100 occasions in the observed dataset.

Please also define the terms "nominal" and "effective".

Page 10, lines 13-14: This phrase is hard to understand. Please consider rephrasing.

Page 12, line 13: "evolution" might be more appropriate that "maturation" in this context.

Page 13, line 1: corr-DSP forecasts

Page 14, line 1: Here, you state that Fig. 6 presents a reliability diagram while in the figure's caption it is labeled as PIT diagram. This is a little bit misleading and might cause confusions even if the type of information is similar to a certain degree. Please confirm or specify the type of figure more detailed.

Page 14, line 11: Here, you state that Fig. 7 presents a PIT histogram while in the figure's caption it is labeled as rank histogram. Is this in line with your explanations in Sect. 4?

Page 14, lines 11-12: Further explanations might improve comprehensibility (e.g., by stating that an equal distribution indicate accurate ensemble forecasts).

Page 14, Figure 6: In my opinion, labeling each row of the diagram by stating the watershed's numbers might be more intuitive (see, e.g., Figure 2). This is also relevant in the case of Figures 7, 8, 9, and 10.

Page 15, line 16: The bias correction is applied for each month. Single events at time scales smaller than one month might by be subject to biases different to the monthly values.

Page 16, line 11: By the way, the term dispersion is often used throughout the manuscript if the variability is overestimated (or underestimated). Variability might be more appropriate as mentioned in line 6 on page 15.

Page 17, line 11: Do you mean corr-DSP when discussing the results of ensemble meteorological forecasts?

Page 17, line 12: Is it possible to prove if the skill is significant or not significant from your analyses? The term significant should be proved by providing statistical measures.

Page 18, line 5-6: Please explain in brief why corr-DSP is less reliable. Is this finding relevant for winter or all seasons? Maybe you can refer to the reliability diagram?

**Technical corrections**

Page 1, line 6: Please add the CRPS (abbreviation) here as it is mentioned later without explanation (cf. line 14)

Page 2, line 32: I would suggest using the singular form of precipitation

Page 8, line 20: distributions (plural)

Page 10, Figure 2: Please add the dimension of the precipitation bias in the color bar.

Page 15, line 15: remain

Page 17, line 11: "is predictable" instead of "are predictable"

Page 18, line 6: originates

---

## Referee Comment (RC2) · Anonymous Referee #2 · 31 Jul 2017

General comment: The paper presents study of potential skill of different meteorological forcing for seasonal forecasting over 10 basins in Quebec that are operationally (short-term) forecasted and economically used for hydro-power production. For these basins in particular, a seasonal forecasting system delivering streamflow volume forecast might be of great potential economic benefits resulting from more effective operation planning. The aim of the study is to compare three methods of seasonal forecasting, namely: a) hydroclimatology (based on simulated streamflow); b) ESP (streamflow simulation based on known initial conditions of the basin and ensemble of historical precipitation and temperature observations); and 3) dynamic hydrological modelling using ECMWF seasonal forecasts of precipitation and temperature. Topic of the paper is fully appropriate for the HESS. Authors present solid introduction and literature

[Figure]

Interactive
comment

review. They use correct methodology that is generally well explained. Results are presented in a clear and understandable manner. The results are probably less optimistic than one might expect when a complex dynamic modelling approach is implemented, especially for a lead times longer than 1 month, however even negative (or not clearly positive) results are worth of publication (I suspect the limited resolution of aggregated observed meteorological data to be one of the factors that contributed to bit fuzzy results.). I recommend accepting the paper after some minor revisions to the paper as proposed bellow.

Authors presents results in more detail for 3 of 10 researched, as they are referred as representing different behaviour of evaluation statistics. For readers, I believe, some more explanation (e.g. on how basins are clustered in this aspect to groups represented by selected basins) would be beneficial. This should also be reflected in the discussion of results (could some physical geographical characteristics be the underlying reason? Do the verification results correlate or not with N-S performance of the hydrological model for these basins?). Authors use simple linear bias correction of ECMWF System4 Forecasts based on differences between forecast mean and observation on a monthly time scale. This method doesn't reflect the ensemble spread of the forecast or the temporal variability of precipitation and temperature within individual months. It would be valuable if authors shortly discuss this issue, in particular, if the bias corrected precipitation and temperature forecasts exhibit ensemble spread over-prediction or under-prediction behaviour (it might have a consequence for interpretation of stream flow and volume forecast results). In general, I would suggest that reasons of a failure of corr-DSP to outperform the ESP beyond 1 month lead time are further investigated and discussed.

Specific comments: p. 1 lines 13 to 16 – I am afraid that the wording of abstract doesn't reflect properly results presented in the paper itself. p. 11 line 4 "... of bias corrected forecasts. The raw ensemble..." p. 13 line 16 Authors state that "in general, corr-DSP outperforms ESP for the 1-month lead-time for watershed 5 and 7." Just bye eye

control of figure 5, I haven't that intention especially as for basin 5 the ESP performs much better for winter period. p. 15 line 9 "...(a) ESP and (b) corr-DSP..." p. 16-19 figures 8 to 10 present 1, 2 and 3 months lead times of spring freshet forecasts. This is defined as (for majority of basins) period from April 1st to June 30th. Does it mean that the 1-month lead-time is forecast issued on March 1st (etc.). Please note that in fig. 11 this is obviously the case as the 0 months lead time is also included. More description of graphical symbols in fig. 8 to 11 should be provided too. p. 18, line 2-3 consider to use "monthly flow volume" instead of "monthly volume" p. 18, line 6 Authors use term "dispersion" throughout the paper, e.g. "this possibly originate from bias propagation or dispersion issues." However, I am afraid that the meaning of "dispersion" is not clear and needs some correction (e.g. ensemble spread of meteorological inputs, variability of...).
* * *

---

## Referee Comment (RC3) · Anonymous Referee #3 · 14 Aug 2017

General comments

The paper is well written, and technically and scientifically sound. Applied methods and data used are well described, and results are presented in a concise and clear way. The paper uses methodologies and results from previous research. The main contribution is the verification of bias-corrected ECMWF System 4 forecasts for hydrological forecasting in Quebec, Canada. This supplements, and to a large degree confirms, previous verification studies in other regions.

Detailed comments

1. Page 7, line 28-30. The procedure for deriving catchment average precipitation and temperature is not that clear. Why is it necessary to first downscale ECMWF forecasts

and then aggregate over a catchment?

2. Page 9, line 18-20. Repetition. Described earlier.

3. Page 10, line 12-13. Both precipitation and temperature are bias-corrected.

4. Page 13, line 16. General performance of watersheds 5 and 7 described is not clear from Fig. 5.

5. Page 14, Figure 6. Reliability diagrams are shown, I expect.

6. Page 15, Figure 7. PIT histograms and not rank histograms, I expect.

7. Page 16, line 1-2. The problem of underdispersion of the bias-corrected ensemble could be elaborated. There is a general overestimation of precipitation cf. Fig. 2. In this case, linear scaling will produce a bias-corrected ensemble with smaller dispersion than the raw ensemble.

---

## Author Comment (AC1) · 6 Sep 2017

**General comments**
In this research article, the authors present a detailed study on the predictive skill of hydrological ensemble forecasts in 10 watersheds in Québec, Canada. Different methods are employed which are subject to different degrees of complexity. Among these methods, a simple application of historical streamflow data is seen as benchmark for more complex approaches. The second method, the ESP approach is based on historical meteorological data and accounts for initial conditions in each forecast. The initialization using known system states is also relevant in the third approach, a dynamical seasonal forecast method, in which meteorological forcing is obtained from bias-corrected climate model

forecasts (ECMWF's System4). Given a lead time of 1 month, the dynamical approach provides improved skill in terms of Continuous Ranked Probability Score (CRPS), while for longer lead times the predictive skill is similar to the corresponding ensemble forecasts using ESP. For the period of snowmelt in spring, the CRPS is lowest (best) in the case of ESP and the dynamical approach. In some watersheds, however, the first method which provides forecasts using historical streamflow data performs best. This comparison highlights the fact that the predictability is low in some watersheds. The study is interesting, the results are promising and the paper fits very well into both the special issue on "Sub-seasonal to seasonal hydrological forecasting" in particular and HESS in general. The methodology is comprehensively presented and the results are discussed in a balanced way. Related work and relevant references are mentioned and acknowledged. Especially the assessment of added value provided by each increased level of complexity (using streamflow data only $->$ ESP $->$ dynamical forecasts) is very useful. Another important point is that the paper presents a specific case study in which operational forecasts have already been issued and new methods are going to be implemented. This might be relevant for other forecasting centers. However, in my opinion, the paper needs a few minor revisions and technical corrections. It's my impression that the section on reliability seems to be detached to a certain degree given that the findings from this analysis are not really considered in the summary. Moreover, the paper would benefit from some additional explanations that might improve comprehensibility. Please find my suggestions below.

Response :
First of all, we would like to thank you for your detailed review and constructive comments. All the technical corrections as well as the specific comments number 1, 4, 6, 13 and 14 have already been integrated in a revised version of the manuscript. Moreover, answers and clarifications for the other specific comments are detailed

below.

**Specific comments :**

1. **Page 1, line 9: the abbreviation "corr-DSP" is not explained in this context and might be omitted here**
Response : The abbreviation will be omitted in the new version of the manuscript.

2. **Page 1, line 9: Would it make sense to point out that "Simulated streamflow computed using observed meteorological data is used as benchmark."?**
Response: Yes, it would make sense. The sentence "Simulated streamflows are used as observations" will be replaced by "Simulated streamflow computed using observed meteorological data is used as benchmark" in the revised version of the manuscript

3. **Page 2, lines 12-28: In this section, historical streamflow prediction (HSP) and extended ensemble streamflow prediction (ESP) are presented. In my opinion, some additional explanations might be helpful in this context. You could explain that using HSP is in general possible without using a hydrological model, even though, in particular, you involve the output of a model in your specific case study. ESP, in contrast, does require a hydrological model in order to improve forecasts through explicitly incorporating initial states in the forecasts. The relevance of using hydrological models, as already pointed out, might be helpful in the process of understanding the different methods you apply.**
Response : Thank you for bringing this to our attention. The relevance of hydrological models will be further detailed in this section in the revised version of the manuscript. The differences between HSP and ESP in that regard will also be explained more clearly.

4. **Page 3, line 2: I am not sure whether "questioning" is the appropriate verb in this context. As far as I know it would make sense if you have reason to doubt the usefulness. Instead, using "assessing" might be a better option.**
Response: We agree with the suggested modification. "Questioning" was replaced by "assessing" in the revised version of the manuscript.

5. **Page 3, line 14: Please add appropriate references of the DEMETER project and also explain the project's acronym.**
Response : The DEMETER acronym stands for 'Development of a European Multimodel Ensemble system for seasonal to inTERannual prediction'. The corresponding reference is:
Palmer, T., Doblas-Reyes, F., Hagedorn, R., Alessandri, A., Gualdi, S., Andersen, U., Feddersen, H., Cantelaube, P., Terres, J., Davey, M., et al.: Development of a European multimodel ensemble system for seasonal-to-interannual prediction (DEMETER), Bulletin of the American Meteorological Society, 85, 853–872, 2004. The definition of the acronym as well as the above mentionned reference will be added in the revised version of the manuscript.

6. **Page 5, Table 1: Please add mean temperature and mean streamflow if easily available. As mean precipitation is indicated, averages of temperature and streamflow might gain insight into the climate characteristics.**
Response: Mean temperature and mean streamflow are indeed easily available and will be added in Table 1 of the revised version of the manuscript.

7. **Page 5, line 13-14: Does this mean that short-term forecasts are extended by the ensembles generated using ESP? Please consider rephrasing.**
Response : Yes, in the operational forecasting system mentioned in the manuscript, the short-term forecasts are extended by ESP. This will be clearly mentioned in the manuscript.

8. **Page 6, lines 27-28: Do the Nash-Sutcliffe values are computed using daily**

**time series?**
Response : Yes. This precision will be added in the revised version of the manuscript.

9. **Page 7, line 18: Forecasts are also computed using one day time steps?**
Response : Yes. Meteorological ensemble forecasts are really computed for 6-hour time steps. However, for this study, forecasts were only available at daily time steps from 0Z to 0Z. Hydrological forecasts are computed at daily time steps. However, hydrological observations are only available at daily time step between 05Z and 05Z. A monthly aggregation of the different variables was chosen for verification purpose in order to limit the impact of the lag between meteorological and hydrological forecasts. Those precisions will be added in the new version of the manuscript.

10. **Page 8, line 6: Please indicate why the number of forecasts amounts to 420. 35 years x 3 months x 4 seasons?**
Response : The total number of forecasts available for verification is 420 because one forecast is emitted the 1st of each month between 1981 and 2014. Consequently, we have 35 years x 12 months to assess the performance of the forecasting system. However, both meteorological and streamflow observations are not available after the 31th of December 2015, lead-time 2 to 7 counts 419 to 413 forecast-observation pairs for the verification. Those precisions will be included in the revised version of the manuscript.

11. **Page 8, line 23: Is the term "confidence interval" really correct in this context? As far as I understand, say we consider a reliable forecast of a specific event, a probability of 95% should at best also refer to 95 out of 100 occasions in the observed dataset. Please also define the terms "nominal" and "effective".**
Response : We propose to add the following sentence in the revised manuscript

to define the terms 'nominal' and 'effective' as well as to clarify the use of the term 'confidence interval': *The reliability diagram diagnostic tool compares the observed coverage frequency (effective, $1 - \hat{\alpha}$) with the corresponding theoretical confidence levels (nominal, $1 - \alpha$) of predictive confidence intervals calculated from ensemble forecasts. Of course, if forecasts are reliable, these values $1 - \hat{\alpha}$ and $1 - \alpha$ should be equal for any confidence level.*

12. **Page 10, lines 13-14: This phrase is hard to understand. Please consider rephrasing.**
    Response : We suggest replacing the sentence 'A leave-one-year-out procedure is used, which consists in excluding the forecast to correct from the bias evaluation process.' by '*A leave-one-year-out procedure is used to calculate the bias and correct the forecast. This consists in calculating the bias based on available past forecasts issued on the same month excluding the month under correction.*'

13. **Page 12, line 13: "evolution" might be more appropriate that "maturation" in this context.**
    Response: We agree. The revised version of the manuscript will be corrected according to your suggestion.

14. **Page 13, line 1: corr-DSP forecasts**
    Response: This will be corrected in the revised version of the manuscript.

15. **Page 14, line 1: Here, you state that Fig. 6 presents a reliability diagram while in the figure's caption it is labeled as PIT diagram. This is a little bit misleading and might cause confusions even if the type of information is similar to a certain degree. Please confirm or specify the type of figure more detailed.**
    Response : We apologize for this typo. The title of Fig. 6 has already been corrected in the revised version of the manuscript.

16. **Page 14, line 11: Here, you state that Fig. 7 presents a PIT histogram while in the figure's caption it is labeled as rank histogram. Is this in line with your explanations in Sect. 4?**
Response : You are right, Figure 7 should be entitled 'PIT diagram' instead of 'Rank histogram'. This will be corrected in the revised version of the manuscript. As mentioned in section 4, PIT histograms and rank histograms are equivalent in terms of interpretation. However, Fig. 7 really represents a PIT histogram.

17. **Page 14, lines 11-12: Further explanations might improve comprehensibility (e.g., by stating that an equal distribution indicate accurate ensemble forecasts).**
Response : Further explanations about the interpretation (flat, bias and over/under-dispersive cases) of the PIT histogram will be added in the revised version of the manuscript to help the interpretation of Figure 7.

18. **Page 14, Figure 6: In my opinion, labeling each row of the diagram by stating the watershed's numbers might be more intuitive (see, e.g., Figure 2). This is also relevant in the case of Figures 7, 8, 9, and 10.**
Response : Yes. The watershed's numbers will be added directly on the rows of the figures 6, 7, 8, 9 and 10 instead of featuring only in the figures' labels.

19. **Page 15, line 16: The bias correction is applied for each month. Single events at time scales smaller than one month might by be subject to biases different to the monthly values.**
Response : We completely agree. This is certainly a limit of the chosen bias correction method (which has the advantage of being simple, but it is possible that in this case a more sophisticated bias correction method would be worth the additional complexity). This issue will be discussed in the revised version of the manuscript.

20. **Page 16, line 11: By the way, the term dispersion is often used throughout**
**the manuscript if the variability is overestimated (or underestimated). Variability might be more appropriate as mentioned in line 6 on page 15.**

Response : The same issue has been addressed by referee #2. Throughout the manuscript, *Dispersion* refers directly to the spread of a single ensemble forecast, namely the variability of the members. This definition of the term *dispersion* will be added in the revised manuscript. We use the term *variability* to characterize the variation in a data set, such as the CRPS of different forecasts.

21. **Page 17, line 11: Do you mean corr-DSP when discussing the results of ensemble meteorological forecasts?**

    Response : No. The denomination corr-DSP refers to streamflow forecasts only. The general denomination 'ensemble meteorological forecasts' refers to uncorrected temperature and precipitation forecasts. Bias corrected ensemble meteorological forecasts are the forecasts used to produce corr-DSP (by passing them to the hydrological model).

22. **Page 17, line 12: Is it possible to prove if the skill is significant or not significant from your analyses? The term significant should be proved by providing statistical measures.**

    Response : We agree. It would be feasible to compute approximate confidence intervals for the CRPS using a bootstrap procedure. These intervals could then be used to add some more formal indications about the significance of the verification results. This will be done for the revised version of the manuscript. The remaining of the manuscript will also be verified to make sure that there are no other instances.

23. **Page 18, line 5-6: Please explain in brief why corr-DSP is less reliable. Is this finding relevant for winter or all seasons? Maybe you can refer to the reliability diagram?**

    Response : As shown in Figure 6 (reliability diagrams), the reliability for corr-

[Figure]

DSP varies in with lead-time, season and watershed. The causes of this lack of reliability are more visible in Figure 7. In some cases, such as 1a) or 2d), biases are still present. In other cases, under-dispersive behaviors are observed such as in cases 1b) or 2c). This inadequate forecast uncertainty representation behavior could be caused by the bias correction which may have reduced the dispersion of the precipitation forecasts. Those explanations will be included in the revised version of the manuscript.

**Technical corrections :**
**Page 1, line 6: Please add the CRPS (abbreviation) here as it is mentioned later without explanation (cf. line 14)**
**Page 2, line 32: I would suggest using the singular form of precipitation**
**Page 8, line 20: distributions (plural)**
**Page 10, Figure 2: Please add the dimension of the precipitation bias in the color bar.**
**Page 15, line 15: remain**
**Page 17, line 11: "is predictable" instead of "are predictable"**
**Page 18, line 6: originates**

Response: We agree with all the suggested technical corrections and we thank you for pointing them out. They have already been included in the revised version of the manuscript that we are preparing.

---

## Author Comment (AC2) · 6 Sep 2017

**General comment: The paper presents study of potential skill of different meteorological forcing for seasonal forecasting over 10 basins in Quebec that are operationally (short-term) forecasted and economically used for hydro-power production. For these basins in particular, a seasonal forecasting system delivering streamflow volume forecast might be of great potential economic benefits resulting from more effective operation planning. The aim of the study is to compare three methods of seasonal forecasting, namely: a) hydroclimatology (based on simulated streamflow); b) ESP (streamflow simulation based on known initial conditions of the basin and ensemble of historical precipitation and temperature observations); and 3) dynamic hydrological modelling using**

[Figure]

ECMWF seasonal forecasts of precipitation and temperature. Topic of the paper is fully appropriate for the HESS. Authors present solid introduction and literature review. They use correct methodology that is generally well explained. Results are presented in a clear and understandable manner. The results are probably less optimistic than one might expect when a complex dynamic modelling approach is implemented, especially for a lead times longer than 1 month, however even negative (or not clearly positive) results are worth of publication (I suspect the limited resolution of aggregated observed meteorological data to be one of the factors that contributed to bit fuzzy results.). I recommend accepting the paper after some minor revisions to the paper as proposed bellow. Authors presents results in more detail for 3 of 10 researched, as they are referred as representing different behaviour of evaluation statistics. For readers, I believe, some more explanation (e.g. on how basins are clustered in this aspect to groups represented by selected basins) would be beneficial. This should also be reflected in the discussion of results (could some physical geographical characteristics be the underlying reason? Do the verification results correlate or not with N-S performance of the hydrological model for these basins?). Authors use simple linear bias correction of ECMWF System4 Forecasts based on differences between forecast mean and observation on a monthly time scale. This method doesn't reflect the ensemble spread of the forecast or the temporal variability of precipitation and temperature within individual months. It would be valuable if authors shortly discuss this issue, in particular, if the bias corrected precipitation and temperature forecasts exhibit ensemble spread over-prediction or under-prediction behaviour (it might have a consequence for interpretation of stream flow and volume forecast results). In general, I would suggest that reasons of a failure of corr-DSP to outperform the ESP beyond 1 month lead time are further investigated and discussed.

Response :

Thank you very much for reviewing the manuscript and providing comments. We agree that the result of this study are not as clearly positive as expected for the performance of ECMWF's System4, especially for lead-times longer than one month. As you mentioned, different reasons can limit the skill of seasonal forecasts, such as the spatial resolution of both observations and forecasts (grid), as well as the choice of a particular bias correction method. The clustering of the watersheds chosen for results presentation in the article will be detailed following your suggestion. Actually, during our study we did not find any clear relationship or pattern between geographical location or characteristics of the watersheds and forecasts' performance. However, the link between N-S performance and the forecasts performance have not been examined. We will examine this aspect in further details and add the information in the revised version of the manuscript. Concerning the time scale in the application of the bias correction method, we agree that it could also be further discussed. Reviewer #1 also pointed out this element.

We agree that others explanations to address the lack of performance for lead-times longer than 1-month will also be investigated and further discussed.

Finally, we would like to thank you for highlighting some technical issues. Answers and clarifications regarding your other comments and suggestions are detailed below each specific comments you outlined.

**Specific comments:**

1. **p. 1 lines 13 to 16 – I am afraid that the wording of abstract doesn't reflect properly results presented in the paper itself.**
   Response: Though, we agree that some nuances can be added, we do not fully agree with your recommendation. The 1st sentence of the highlighted lines is *For the 1-month lead-time, a gain exists for almost all watersheds during winter, summer and fall.*. This sentence is based on the results presented in the bottom row of Figure 5, where the performance of corr-DSP is compared with that of ESP. Except for watershed number 5 during winter, all skill scores are indeed positive for winter, summer and fall, which indicate a gain in performance when using corr-DSP instead of ESP. We exclude spring because the results are too contrasted over the different watersheds. The second sentence is *However, volume forecasts performance for spring is close to the performance of ESP.* This sentence is still based on the bottom row of Figure 5. It reflects the fact that for watersheds 2, 4 and 10, the CRPSS is really close to 0. Then, for watersheds 1, 3, 6, 8 and 9, the performance is still close to 0 in favor of ESP or DSP. There is skill only for watersheds 5 and 7 during this season (spring). The third sentence *For longer lead-times, results are mixed and the CRPS skill score is close to 0 in most cases.* is still based on the same figure. Even if the CRPSS is close to 0, the color scale shows that if a preference is given, it is, in most cases, in favor of ESP. This precision could easily be added in the abstract to make it more precise, according to your comment. The last sentence *Bias-corrected ensemble meteorological forecasts appear to be an interesting source of information for hydrological forecasting.* could indeed benefit from a reformulation. Corr-DSP is interesting compared to the use of streamflow climatology. Moreover, compared to ESP, the added value of Corr-DSP is mostly visible for the 1st month. We propose to rewrite the last sentences of the abstract to include the above mentioned nuances as : *For the 1-month lead-time, a gain exists for almost all watersheds during winter, summer and fall. However, volume forecasts performance for spring varies from one watershed to another. For most of them, the performance is close to the performance of ESP. For longer lead-times, the CRPS skill score is mostly in favor of ESP, even if for many watersheds, ESP and corr-DSP have comparable skill. Bias-corrected ensemble meteorological forecasts appear to be an interesting source of information for hydrological forecasting for lead-times up to 1-month. They could also complement ESP for longer lead-times.*

2. **p. 11 line 4 "... of bias corrected forecasts. The raw ensemble ..."**
Response: We apologize for this, it has already been corrected in the revised version of the manuscript that is currently in preparation.

3. **p. 13 line 16 Authors state that "in general, corr-DSP outperforms ESP for the 1-month lead-time for watershed 5 and 7." Just bye eye control of figure 5, I haven't that intention especially as for basin 5 the ESP performs much better for winter period.**
Response : This is a mistake as we clearly see that the watershed 5 is not a good example in this sentence. The sentence *"in general, corr-DSP outperforms ESP for the 1-month lead-time for watershed 5 and 7."* will be replaced by *"In general, corr-DSP outperforms ESP for the 1-month lead-time, with some exceptions such as watershed number 5 in winter or watersheds number 3 and 9 during the spring."*

4. **p. 15 line 9 "...(a) ESP and (b) corr-DSP ..."**

Response: This has been corrected in the revised version of the manuscript, thank you for pointing this out.

5. **p. 16-19 figures 8 to 10 present 1, 2 and 3 months lead times of spring freshet forecasts. This is defined as (for majority of basins) period from April 1st to June 30th. Does it mean that the 1-month lead-time is forecast issued on March 1st (etc.). Please note that in fig. 11 this is obviously the case as the 0 months lead time is also included. More description of graphical symbols in fig. 8 to 11 should be provided too.**

Response: In figures 8 to 10, the 1-month lead-time corresponds to the forecast issued on the 1st day of the spring freshet (namely on the 1st of April for the majority of basins), for the following month. This is obviously incoherent with

[Figure]

Figure 11. We thank you very much for pointing out this mistake. Lead-time is the delay between the date of emission of the forecast and the end of the validity period. We propose to add this definition to the paper. Moreover, we will make sure that all results and figure labels in the revised version of the manuscript are coherent with this definition of lead-time.

6. **p. 18, line 2-3 consider to use "monthly flow volume" instead of "monthly volume"**
   Response: We will, thank you.

7. **p. 18, line 6 Authors use term "dispersion" throughout the paper, e.g. "this possibly originate from bias propagation or dispersion issues." However, I am afraid that the meaning of "dispersion" is not clear and needs some correction (e.g. ensemble spread of meteorological inputs, variability of ...).**
   Response: The same issue has been addressed by referee #1. Throughout the manuscript, *Dispersion* refers directly to the spread of a single ensemble forecast, namely the variability of the members. This definition of the term 'dispersion' will be added in the revised manuscript.

---

## Author Comment (AC3) · 6 Sep 2017

**General comments**
**The paper is well written, and technically and scientifically sound. Applied methods and data used are well described, and results are presented in a concise and clear way. The paper uses methodologies and results from previous research. The main contri- bution is the verification of bias-corrected ECMWF System 4 forecasts for hydrological forecasting in Quebec, Canada. This supplements, and to a large degree confirms, previous verification studies in other regions.**
Response : We would like to thank you for your review and comments. Answers and clarifications for the detailed comments are detailed below.

[Figure]

**Detailed comments**
**1. Page 7, line 28-30. The procedure for deriving catchment average precipitation and temperature is not that clear. Why is it necessary to first downscale ECMWF forecasts and then aggregate over a catchment?**
Response : The original resolution of ECMWF System4 forecast grid is 0.7 degrees. This is too coarse to used at watershed scale, as some watersheds have no point or only 1-2 grid points inside their boundaries. In order to have more points inside the watershed boundaries and ensure that the average precipitation is not biased by a local storm, the original grid was interpolated on a 0.1° grid.

**2. Page 9, line 18-20. Repetition. Described earlier.**
Response : Although we agree that the information is repeated, it was made on purpose and we would prefer to leave the sentence there. Indeed, the denominations 'DSP' and 'HSP' are not conventional and we wanted to remind the reader about their meaning at this point in the manuscript. Reviewer 1's 21st specific comment (about Page 17, line 11) also indicates that the denominations used in the manuscript can perhaps be confusing, so we would really prefer to keep this repetition.

**3. Page 10, line 12-13. Both precipitation and temperature are bias-corrected.**
Response : Yes, they are both bias-corrected. Indeed, the way the sentence is formulated in the present version of the manuscript does not include temperature. This will be specified in the improved version of the manuscript.

**4. Page 13, line 16. General performance of watersheds 5 and 7 described is not clear**
Response : This issue was also raised by Reviewer 2 (his or her 3rd specific comment).

This is a mistake, which we will correct. In fact, we clearly see that watershed 5 is not a good example in this sentence. The sentence *"in general, corr-DSP outperforms ESP for the 1-month lead-time for watershed 5 and 7."* will be replaced by *"In general, corr-DSP outperforms ESP for the 1-month lead-time, with some exceptions such as watershed number 5 in winter or watersheds number 3 and 9 during the spring."*

**6. Page 15, Figure 7. PIT histograms and not rank histograms, I expect.**
Response : Thank you for highlighting this mistake, which was also pointed out by Reviewer 1. This has already been corrected in the revised version of the manuscript currently in preparation.

**7. Page 16, line 1-2. The problem of underdispersion of the bias-corrected ensemble could be elaborated. There is a general overestimation of precipitation cf. Fig. 2. In this case, linear scaling will produce a bias-corrected ensemble with smaller dispersion than the raw ensemble.**
Response : We agree. The linear scaling method modifies the dispersion of precipitation forecasts and can influence the dispersion of streamflow forecasts. This will be further discussed in the revised version of the manuscript and the elements you mention will be specifically mentioned.

――――――――――――――――――

---

## Author Response (AR1)

First of all, we would like to thank the editor for considering our paper worth of interest. Second, the constructive comments of the three reviewers helped us to improve the original manuscript. We tried to address their questions and comments as accurately as possible to improve the manuscript. We hope our answers meet their expectations. Please note that all page and line numbers refer to the track-change version of the manuscript.

**1  Response to Reviewer 1**

**General comments**
**In this research article, the authors present a detailed study on the predictive skill of hydrological ensemble forecasts in 10 watersheds in Québec, Canada. Different methods are employed which are subject to different degrees of complexity. Among these methods, a simple application of historical streamflow data is seen as benchmark for more complex approaches. The second method, the ESP approach is based on historical meteorological data and accounts for initial conditions in each forecast. The initialization using known system states is also relevant in the third approach, a dynamical seasonal forecast method, in which meteorological forcing is obtained from bias-corrected climate model forecasts (ECMWF's System4). Given a lead time of 1 month, the dynamical approach provides improved skill in terms of Continuous Ranked Probability Score (CRPS), while for longer lead times the predictive skill is similar to the corresponding ensemble forecasts using ESP. For the period of snowmelt in spring, the CRPS is lowest (best) in the case of ESP and the dynamical approach. In some watersheds, however, the first method which provides forecasts using historical streamflow data performs best. This comparison highlights the fact that the predictability is low in some watersheds. The study is interesting, the results are promising and the paper fits very well into both the special issue on "Sub-seasonal to seasonal hydrological forecasting" in particular and HESS in general. The methodology is comprehensively presented and the results are discussed in a balanced way. Related work and relevant references are mentioned and acknowledged. Especially the assessment of added value provided by each increased level of complexity (using streamflow data only − > ESP − > dynamical forecasts) is very useful. Another important point is that the paper presents a specific case study in which operational forecasts have already been issued and new methods are going to be implemented. This might be relevant for other forecasting centers. However, in my opinion, the paper needs a few minor revisions and technical corrections. It's my impression that the section on reliability seems to be detached to a certain degree given that the findings from this analysis are not really considered in the summary. Moreover, the paper would benefit from some additional explanations that might improve comprehensibility. Please find my suggestions below.**

Response :
Thank you once again for your detailed review and constructive comments. CRPS is used to assess the overall performance of the forecasting system. Reliability diagram and PIT histogram have been used as diagnostic tools to understand the lack of performance of the ensemble forecasts. The conclusion is that even if the reliability is quite good, some under-dispersive behavior or bias are observed in some cases. As the number of event are limited for the verification, these punctual problems can penalize strongly the CRPS of corr-DSP. Some part of the summary has been rewritten to take into account your remark as follows at Page 1, line 18-20 : "Corr-DSP appears quite reliable but, in some cases, under-dispersion or bias is observed. A more complex bias correction method should be further investigated to remedy this weakness and take more advantage of the ensemble forecasts produced by the climate model." Answers and clarifications for the other specific comments are detailed below.

**Specific comments :**

1. **Page 1, line 9: the abbreviation "corr-DSP" is not explained in this context and might be omitted here**
   Page 1, line 9 : The explicit meaning of the abbreviation is not explained here, as well as ESP at page 1, line 8. However, these abbreviations are useful here as we refer to them in the rest of the abstract. The word "corr-DSP" has been kept in the manuscript.

2. **Page 1, line 9: Would it make sense to point out that "Simulated streamflow computed using observed meteorological data is used as benchmark."?**

   Page 1, lines 10-11: The sentence "Simulated streamflows are used as observations" has been replaced by "Simulated streamflow computed using observed meteorological data is used as benchmark".

3. **Page 2, lines 12-28: In this section, historical streamflow prediction (HSP) and extended ensemble streamflow prediction (ESP) are presented. In my opinion, some additional explanations might be helpful in this context. You could explain that using HSP is in general possible without using a hydrological model, even though, in particular, you involve the output of a model in your specific case study. ESP, in contrast, does require a hydrological model in order to improve forecasts through explicitly incorporating initial states in the forecasts. The relevance of using hydrological models, as already pointed out, might be helpful in the process of understanding the different methods you apply.**

Page 2, lines 27-28 : The sentence "Contrary to HSP, ESP require the use of a calibrated hydrological model to produce hydrological forecasts" has been added.

4. **Page 3, line 2: I am not sure whether "questioning" is the appropriate verb in this context. As far as I know it would make sense if you have reason to doubt the usefulness. Instead, using "assessing" might be a better option.**

Page 3, line 14 : We agree and the word "questioning" has been replaced by "assessing".

5. **Page 3, line 14: Please add appropriate references of the DEMETER project and also explain the project's acronym.**

Page 3, line 26-27 : The signification of the DEMETER acronym "Development of a European Multimodel Ensemble system for seasonal to inTERannual prediction" has been added.

Also on page 3, line 27, the following reference has been cited according to your suggestion: "Palmer, T., Doblas-Reyes, F., Hagedorn, R., Alessandri, A., Gualdi, S., Andersen, U., Feddersen, H., Cantelaube, P., Terres, J., Davey, M., et al.: Development of a European multimodel ensemble system for seasonal-to-interannual prediction (DEMETER), Bulletin of the American Meteorological Society, 85, 853–872, 2004"

6. **Page 5, Table 1: Please add mean temperature and mean streamflow if easily available. As mean precipitation is indicated, averages of temperature and streamflow might gain insight into the climate characteristics.**

Page 6, Table 1 : Information about the mean temperature and mean streamflow have been added. Also, the labels of the columns for temperature data have been changed to clarify the content. The content have also been updated. Consequently, small changes can appear compared to the previous version of the manuscript.

7. **Page 5, line 13-14: Does this mean that short-term forecasts are extended by the ensembles generated using ESP? Please consider rephrasing.**

Page 6, line 1-10 and page 7, line 1-20 : The first paragraph of the operational system description has been reformulated as follows: "The current operational streamflow forecasting system at Hydro-Québec relies on Extended Streamflow Prediction (ESP, Day, 1985) and can be divided into three distinct stages. In the first stage, an analog approach (e.g. Marty et al., 2012) based upon deterministic meteorological forecasts from Environment and Climate Change Canada is used to produce short-term meteorological forecasts. The definition of "short-term" is not fixed. The lead-time depends on watersheds and meteorological events. On average, five to seven days ahead forecasts are produced using this method. The second stage aims to produce seasonal forecasts. Observed precipitation and temperature for previous years are considered as plausible future scenarios. Hence, archived observed meteorological conditions for all previous years (since 1950) form an ensemble. These analog-based meteorological ensembles are used to extend the short-term forecasts obtained in the first stage. These scenarios are then fed to a lumped conceptual hydrological model (described below) to obtain hydrological ensemble forecasts. Lastly, the third and last stage begins when the influence of initial conditions becomes negligible. Observed streamflow for the same Julian day of each available year in the database are then considered as equiprobable long-term forecasts (Historical Streamflow Predictions, see Introduction). The appropriate moment to shift from ESP to HSP is fixed by the forecaster and varies between watersheds. Note that Hydro-Quebec is currently improving its forecasting system by integrating ensemble weather forecasts with statistical post-processing for short-term forecasting, and by developing a weather generator for medium-term forecasting. This new system is expected to become operational in 2018."

8. **Page 6, lines 27-28: Do the Nash-Sutcliffe values are computed using daily time series?**

Page 8, line 7 : The sentence "The Nash-Sutcliffe efficiencies (NSE) ranges from 0.30 to 0.86 for the 1981-2015 period." has been changed for "The Nash-Sutcliffe efficiencies (NSE) based on daily streamflow data range from 0.30 to 0.86 for the 1981-2015 period" in order to be more specific.

9. **Page 7, line 18: Forecasts are also computed using one day time steps?**

Response : Yes. Meteorological ensemble forecasts are really computed for 6-hour time steps. However, for this study, forecasts were only available at daily time steps from 0Z to 0Z. Page 8, lines 32-33 : To clarify, the sentence "Ensemble forecasts are computed at finer time-steps than one day but are available only at daily time-step from 00Z to 00Z for this study." has been added. Furthermore, hydrological forecasts are computed at daily time steps. However, hydrological observations are only available at daily time step between 05Z and 05Z as mentioned in Section 2. In addition, the following sentence has been added at page 9, lines 19-22: "Both meteorological and hydrological forecasts are available at daily time steps. However, as mentioned in section 2 and 3, a lag exists between daily forecasts and observations. A monthly aggregation of the different variables is performed for verification, in order to limit the impact of the lag between forecasts (meteorological and hydrological) and observations."

10. **Page 8, line 6: Please indicate why the number of forecasts amounts to 420. 35 years x 3 months x 4 seasons?**

Page 9, lines 32-33 : The sentence "Because both meteorological and streamflow observations are not available after the 31th of December 2015, 2 to 7-months lead-time have 419 to 413 forecast-observation pairs for the verification, respectively." has been added. In addition, on page 10, line 7-9, the sentence "For one season and one lead-time, each set of verification comprises 105 monthly ensemble forecast-observation pairs" was modified for "For one season and one lead-time, each set of verification comprises around between 100 and 105 monthly ensemble forecast-observation pairs."

11. **Page 8, line 23: Is the term "confidence interval" really correct in this context? As far as I understand, say we consider a reliable forecast of a specific event, a probability of 95% should at best also refer to 95 out of 100 occasions in the observed dataset. Please also define the terms "nominal" and "effective".**

Page 10, lines 19-22: In order to define the terms 'nominal' and 'effective' as well as to clarify the use of the term 'confidence interval', the sentence "Confidence intervals computed from reliable forecasts should be in agreement with their definition: the 95% confidence interval, for instance, must include on average 95 observations out of 100. For each nominal confidence level probability from 0.1 to 0.9, the effective frequency of the observation occurrence in the given nominal interval is calculated. Then, the effective frequencies are plotted against the nominal confidence level probability." now reads "The reliability diagram diagnostic tool compares the observed coverage frequency (effective, $1 - \hat{\alpha}$) with the corresponding theoretical confidence levels (nominal, $1 - \alpha$) of predictive confidence intervals calculated from ensemble forecasts. Of course, if forecasts are reliable, these values $1 - \hat{\alpha}$ and $1 - \alpha$ should be equal for any confidence level."

12. **Page 10, lines 13-14: This phrase is hard to understand. Please consider rephrasing.**

Page 12, lines 21-23 : The sentence "A leave-one-year-out procedure is used, which consists in excluding the forecast to correct from the bias evaluation process" was modified to improve clarity and now reads "A leave-one-year-out procedure is used to calculate bias and correct the forecasts. This consists in calculating the bias based on available forecasts issued on the same month, excluding the month under correction".

13. **Page 12, line 13: "evolution" might be more appropriate that "maturation" in this context.**

Page 14, line 5: We agree. The word "maturation" has been replaced by "evolution".

14. **Page 13, line 1: corr-DSP forecasts**

Page 14, line 10: Thank you for pointing this out. The word "ensemble forecasts" has been replaced by "corr-DSP".

15. **Page 14, line 1: Here, you state that Fig. 6 presents a reliability diagram while in the figure's caption it is labeled as PIT diagram. This is a little bit misleading and might cause confusions even if the type of information is similar to a certain degree. Please confirm or specify the type of figure more detailed.**

Page 17, Figure 6 : We apologize for this typo. The words "PIT histograms" have been replaced by "Reliability diagrams" in the title of the figure.

16. **Page 14, line 11: Here, you state that Fig. 7 presents a PIT histogram while in the figure's caption it is labeled as rank histogram. Is this in line with your explanations in Sect. 4?**

Page 17, Figure 7 : You are right and "Rank histograms" has been replaced by "PIT histograms" in the title of the figure. As mentioned in section 4, PIT histograms and rank histograms are equivalent in terms of interpretation. However, Fig. 7 really represents a PIT histogram.

17. **Page 14, lines 11-12: Further explanations might improve comprehensibility (e.g., by stating that an equal distribution indicate accurate ensemble forecasts).**

Page 16, lines 26-31 : Further explanations about the interpretation (flat, bias and over/under-dispersive cases) of the PIT histogram has been added as follows: "A flat PIT histogram corresponds to an accurate forecasting system whereas a higher effective frequency on one side of the histogram indicates the presence of bias (asymetric shape). Higher effective frequency in the middle of the PIT histogram is linked with too much dispersion of the ensemble (bell-shape) and, on the contrary, higher effective frequencies on both sides of the histogram is the sign of an under-dispersive behavior (U-shape, the spread of the ensemble is too small for most forecasts)."

18. **Page 14, Figure 6: In my opinion, labeling each row of the diagram by stating the watershed's numbers might be more intuitive (see, e.g., Figure 2). This is also relevant in the case of Figures 7, 8, 9, and 10.**

Pages 17, 19, 20, 21, Figures 6, 7, 8, 9 and 10 : The watershed's numbers have been added directly on the rows of the figures 6 and 7 and the lead-time on the rows of the figures 8, 9 and 10 to your suggestion.

19. **Page 15, line 16: The bias correction is applied for each month. Single events at time scales smaller than one month might be subject to biases different to the monthly values.**

Response: We completely agree. This is certainly a limit of the chosen bias correction method, which has the advantage of being simple, but it is possible that in this case a more sophisticated bias correction method would be worth the additional complexity. Hence, the sentence "Moreover, single events at time scales smaller than one month might by be subject to biases different than the monthly values used for bias correction" has been added on page 18, lines 19-20.

20. **Page 16, line 11: By the way, the term dispersion is often used throughout the manuscript if the variability is overestimated (or underestimated). Variability might be more appropriate as mentioned in line 6 on page 15.**

Response : The same issue has been raised by Reviewer #2. Throughout the manuscript, *Dispersion* refers directly to the spread of a single ensemble forecast, namely the variability of the members. To clarify the meaning of the word *dispersion*, the following sentence has been added on page 10, line 18: "In the following, the term 'dispersion' refers to the spread of the ensemble forecasts" The term 'variability' could also be used but is less frequent than the term 'dispersion.

21. **Page 17, line 11: Do you mean corr-DSP when discussing the results of ensemble meteorological forecasts?**

Page 11, lines 18-19 : To clarify, the sentence "raw and bias-corrected meteorological forecasts will refer to the ensemble meteorological forecasts of the ECMWF System4" has been added.

22. **Page 17, line 12: Is it possible to prove if the skill is significant or not significant from your analyses? The term significant should be proved by providing statistical measures.**

Response : We agree. We computed approximate confidence intervals for the CRPS of corr-DSP and ESP using a bootstrap procedure. These intervals could then be used to add some more formal indications about the significance of the verification results. This has been done for the revised version of the manuscript. We found that the difference between the CRPS of corr-DSP and ESP are not significant for almost all cases. The remaining of the manuscript has also been verified to make sure that there are no other instances. Page 19, line 12 : The word "significant" has been deleted.

Page 21, line 8 : The word 'significant' has been replaced by 'notable'.
Page 21, line 12 : The word 'significant' has been replaced by 'important'.

23. **Page 18, line 5-6: Please explain in brief why corr-DSP is less reliable. Is this finding relevant for winter or all seasons? Maybe you can refer to the reliability diagram?**

Page 20, lines 5-10 : The following sentences have been added: "see Figure 7 for an example, in both meteorological and hydrological forecasts. The lack of skill of corr-DSP can originate from different sources. First, linear scaling is a rather simple bias correction method. It was performed using monthly bias and thus, there is a possibility that biases at smaller temporal scales can remain. Second, as precipitation was originally over-predicted in most cases by System4 (see Figure 2), bias correction results in a reduction of the ensemble spread for precipitation forecasts, and possibly for streamflow forecasts also."

**Technical corrections :**
**Page 1, line 7: Please add the CRPS (abbreviation) here as it is mentioned later without explanation (cf. line 14)**
Page 1-line 6: The word "CRPS" has been added in parenthesis.
**Page 2, line 32: I would suggest using the singular form of precipitation**
Page 3, line 9 : Done!
**Page 8, line 20: distributions (plural)**
Page 10, line 16: Also done.
**Page 10, Figure 2: Please add the dimension of the precipitation bias in the color bar.**
Page 13, Figure 2 has been modified according to your comment.
**Page 15, line 15: remain**
Page 18, line 18: It has been corrected.
**Page 17, line 11: "is predictable" instead of "are predictable"**
Page 19, line 11: The sentence was corrected according to your suggestion.
**Page 18, line 6: originates**
Page 20, line 5: Done!

**2 Response to Reviewer 2**

**General comment: The paper presents study of potential skill of different meteorological forcing for seasonal forecasting over 10 basins in Quebec that are operationally (short-term) forecasted and economically used for hydro-power production. For these basins in particular, a seasonal forecasting system delivering streamflow volume forecast might be of great potential economic benefits resulting from more effective operation planning. The aim of the study is to compare three methods of seasonal forecasting, namely: a) hydroclimatology (based on simulated streamflow); b) ESP (streamflow simulation based on known initial conditions of the basin and ensemble of historical precipitation and temperature observations); and 3) dynamic hydrological modelling using ECMWF seasonal forecasts of precipitation and temperature. Topic of the paper is fully appropriate for the HESS. Authors present solid introduction and literature review. They use correct methodology that is generally well explained. Results are presented in a clear and understandable manner. The results are probably less optimistic than one might expect when a complex dynamic modelling approach is implemented, especially for a lead times longer than 1 month, however even negative (or not clearly positive) results are worth of publication (I suspect the limited resolution of aggregated observed meteorological data to be one of the factors that contributed to bit fuzzy results.). I recommend accepting the paper after some minor revisions to the paper as proposed bellow. Authors presents results in more detail for 3 of 10 researched, as they are referred as representing different behaviour of evaluation statistics. For readers, I believe, some more explanation (e.g. on how basins are clustered in this aspect to groups represented by selected basins) would be beneficial. This should also be reflected in the discussion of results (could some physical geographical characteristics be the underlying reason? Do the verification results correlate or not with N-S performance of the hydrological model for these basins?). Authors use simple linear bias correction of ECMWF System4 Forecasts based on differences between forecast mean and observation on a monthly time scale. This method doesn't reflect the ensemble spread of the forecast or the temporal variability of precipitation and temperature within individual months. It would be valuable if authors shortly discuss this issue, in particular, if the bias corrected precipitation and temperature forecasts exhibit ensemble spread over-prediction or under-prediction behaviour (it might have a consequence for interpretation of stream flow and volume forecast results). In general, I would suggest that reasons of a failure of corr-DSP to outperform the ESP beyond 1 month lead time are further investigated**

**and discussed.**

Response :
Thank you very much for reviewing the manuscript and providing comments. We agree that the result of this study are not as clearly positive as expected for the performance of ECMWF's System4, especially for lead-times longer than one month. As you mentioned, different reasons can limit the skill of seasonal forecasts, such as the spatial resolution of both observations and forecasts (grid), as well as the choice of a particular bias correction method.

The watersheds chosen for results presentation are linked with their CRPS performance. Indeed, watersheds 3 and 5 present opposite behaviors of the CRPSS for the 1-month lead-time during winter and spring. Actually, during our study we did not find any clear relationship or pattern between geographical location or characteristics of the watersheds and forecasts' performance. Moreover, no clear link between N-S performance and the forecasts performance has been noted. Consequently, no mention of these points has been added to the manuscript.

Concerning the time scale in the application of the bias correction method, Reviewer #1 also pointed out this element. Page 20, lines 5-10 : The following sentences have been added: "see Figure 7 for an example, in both meteorological and hydrological forecasts. The lack of skill of corr-DSP can originate from different sources. First, linear scaling is a rather simple bias correction method. It was performed using monthly bias and thus, there is a possibility that biases at smaller temporal scales can remain. Second, as precipitation was originally over-predicted in most cases by System4 (see Figure 2), bias correction results in a reduction of the ensemble spread for precipitation forecasts, and possibly for streamflow forecasts also."

Through the different comments and remarks, we have addressed some of the explanations related to the lack of performance for lead-times longer than 1-month. Further studies are needed to explore which aspects of these potential explanations are the most relevant. Possible future studies could include the comparison of other bias correction methods, the comparison with other products characterized by different resolutions as well as the comparison with gridded meteorological observations instead of watershed average forecasts, among others.

Finally, we would like to thank you for highlighting some technical issues. Answers and modifications regarding your other comments and suggestions are detailed below.

**Specific comments:**

1. **p. 1 lines 13 to 16 – I am afraid that the wording of abstract doesn't reflect properly results presented in the paper itself.**

   Response: Though we agree that some nuances can be added, we do not fully agree with your recommendation.
   Page 1, lines 14-15 : The 1st sentence of the highlighted lines is "For the 1-month lead-time, a gain exists for almost all watersheds during winter, summer and fall." This sentence is based on the results presented in the bottom row of Figure 5, where the performance of corr-DSP is compared with that of ESP. Except for watershed number 5 during winter, all skill scores are indeed positive for winter, summer and fall, which indicate a gain in performance when using corr-DSP instead of ESP. We exclude spring because the results are too contrasted over the different watersheds. This first sentence has been kept as is.
   Page 1, lines 15-16 : The second sentence is "However, volume forecasts performance for spring is close to the performance of ESP." This sentence is still based on the bottom row of Figure 5. It reflects the fact that for watersheds 2, 4 and 10, the CRPSS is really close to 0. Then, for watersheds 1, 3, 6, 8 and 9, the performance is still close to 0 in favor of ESP or DSP. There is skill only for watersheds 5 and 7 during this season (spring). The second sentence was replaced by "However, volume forecasts performance for spring varies from one watershed to another. For most of them, the performance is close to the performance of ESP."
   Page 1, lines 16-18 : The third sentence "For longer lead-times, results are mixed and the CRPS skill score is close to 0 in most cases." is still based on the same figure. Even if the CRPSS is close to 0, the color scale shows that if a preference is given, it is, in most cases, in favor of ESP. This precision could easily be added in the abstract, according to your comment. So, the third sentence has been replaced by "For longer lead-times, the CRPS skill score is mostly in favor of ESP, even if for many watersheds, ESP and corr-DSP have comparable skill."
   Page 1, line 18-20 : The sentence has been added "Corr-DSP appears quite reliable but, in some cases, under-dispersion or bias is observed. A more complex bias correction method should be further investigated to remedy this weakness and take more advantage of the ensemble forecasts produced by the climate model."
   Page 1, lines 20-22 : The last sentence "Bias-corrected ensemble meteorological forecasts appear to be an interesting source of information for hydrological forecasting." could indeed benefit from a reformulation. Corr-DSP is interesting compared to the use of streamflow climatology. Moreover, compared to ESP, the added value of Corr-DSP is mostly visible for the 1st month. The last sentence has been replaced by "Overall, in this study, bias-corrected ensemble meteorological forecasts appear to be an interesting source of information for hydrological forecasting for lead-times up to 1-month. They could also complement ESP for longer lead-times."

2. **p. 11 line 4 "... of bias corrected forecasts. The raw ensemble ..."**

Response: We apologize for this typo.
Page 12, line 27 : The word "with" has been deleted.

3. **p. 13 line 16 Authors state that "in general, corr-DSP outperforms ESP for the 1-month lead-time for watershed 5 and 7." Just bye eye control of figure 5, I haven't that intention especially as for basin 5 the ESP performs much better for winter period.**

Response : This is a mistake as we clearly see that the watershed 5 is not a good example in this sentence. Page 16, lines 8-10 : The sentence "In general, corr-DSP outperforms ESP for the 1-month lead-time for watershed 5 and 7." has been replaced by "In general, corr-DSP outperforms ESP for the 1-month lead-time, with some exceptions such as watershed number 5 in winter or watersheds number 3 and 9 during spring."

4. **p. 15 line 9 "...(a) ESP and (b) corr-DSP ..."**

Page 18, line 9 : The word "and" has been deleted.

5. **p. 16-19 figures 8 to 10 present 1, 2 and 3 months lead times of spring freshet forecasts. This is defined as (for majority of basins) period from April 1st to June 30th. Does it mean that the 1-month lead-time is forecast issued on March 1st (etc.). Please note that in fig. 11 this is obviously the case as the 0 months lead time is also included. More description of graphical symbols in fig. 8 to 11 should be provided too.**

Response: In figures 8 to 10, the 1-month lead-time corresponds to the forecast issued on the 1st day of the spring freshet (namely on the 1st of April for the majority of basins), for the following month. This is obviously incoherent with Figure 11. We thank you for pointing out this mistake. Lead-time is the delay between the date of emission of the forecast and the end of the validity period. We have added this definition in the manuscript.
Page 11, lines 22-26 : The sentence "Moreover, the lead-time refers to the time lag between the emission date of the forecast and the time at which the forecast is valid. For instance, a skill score for the January-February-March season for the 5-month lead-time correspond to the performance of the forecasts issued 5 months earlier, in August, September and October." has been replaced by the sentence "The lead-time is defined herein as the time between the date of emission of the forecast and the end of the validity period of the forecasts. For instance, the 1-month lead-time of the forecast issued on the 1st of January is the monthly volume or temperature of January."
Page 19, 20, 21, 22, respectively Figure 8, 9, 10, and 11 : The lead-times in Figure 7 to 11, as well as their titles, have been changed accordingly to the previous definition.
Page 18 , line 9 : Concerning the last sentence of the comment, we have modified the original sentence, which now reads "Figure 8 to 10 present the boxplots of ensemble forecasts". We assume that boxplots are well-known tools for all readers.
Page 18, lines 10-13 : This sentence has been added "In those figures, the 3-month lead-time forecast corresponds to the forecasts issued at the beginning of the spring freshet period with a 3-month validity period. The 4-month and the 5-month lead-time forecasts are issued respectively 1 and 2 months before the beginning of the spring freshet period, with corresponding 4- and 5-month validity period."

6. **p. 18, line 2-3 consider to use "monthly flow volume" instead of "monthly volume"**

Page 20, line 2 : The word "flow" has been added.

7. **p. 18, line 6 Authors use term "dispersion" throughout the paper, e.g. "this possibly originate from bias propagation or dispersion issues." However, I am afraid that the meaning of "dispersion" is not clear and needs some correction (e.g. ensemble spread of meteorological**

inputs, variability of ...).

Response: The same issue has been raised by Reviewer #1. Throughout the manuscript, *Dispersion* refers directly to the spread of a single ensemble forecast, namely the variability of the members in a single forecast. Page 10, line 18 : To clarify the definition of the term "dispersion", the following sentence has been added "In the following, the term 'dispersion' refers to the spread of the ensemble forecasts."

**3    Response to Reviewer 3**

**General comments**
**The paper is well written, and technically and scientifically sound. Applied methods and data used are well described, and results are presented in a concise and clear way. The paper uses methodologies and results from previous research. The main contri- bution is the verification of bias-corrected ECMWF System 4 forecasts for hydrological forecasting in Quebec, Canada. This supplements, and to a large degree confirms, previous verification studies in other regions.**

Response : We would like to thank you for your review and comments. Answers and modifications for the detailed comments are detailed below.

**Detailed comments**
**1. Page 7, line 28-30. The procedure for deriving catchment average precipitation and temperature is not that clear. Why is it necessary to first downscale ECMWF forecasts and then aggregate over a catchment?**

Page 9, line 9-16: The original paragraph "Those original resolutions are both too coarse for hydrological applications, as only very few grid points fall inside the watersheds delineations. The original grid was thus downscaled to a 0.1 degree grid through linear interpolation in order to obtain multiple grid points for each watershed. Then, since HSAMI is a lumped model, grid points were averaged to aggregate the information at the watershed scale." has been modified as follows "Since HSAMI is a lumped model, meteorological forecasts has to be a single point representative of the meteorological conditions over the watershed. The original resolution is too coarse for hydrological applications, as only very few grid points fall inside the watersheds delineations. The original grid was thus downscaled to a 0.1 degree grid through linear interpolation in order to obtain more points inside the watersheds boundaries. This allows to ensure that points close to the watersheds boundaries contribute to more accurate meteorological forecasts over the watershed. Then, grid points were averaged to aggregate the information at the watershed scale."

**2. Page 9, line 18-20. Repetition. Described earlier.**

Response : Although we agree that the information is repeated, it was made on purpose and we would prefer to leave the sentence there. Indeed, the denominations 'DSP' and 'HSP' are not conventional and we wanted to remind the reader about their meaning at this point in the manuscript. Reviewer 1's 21st specific comment (about Page 17, line 11) also indicates that the denominations used in the manuscript can perhaps be confusing, so we would really prefer to keep this repetition.

**3. Page 10, line 12-13. Both precipitation and temperature are bias-corrected.**

Response : Yes, they are both bias-corrected.
Page 12, lines 19-20 : The original sentence has been completed by "as well as minimal and maximal temperatures".
Page 12, lines 17-19 : Moreover, we found that the previous sentence was not clear. It was thus modified and now reads "As mentioned above, Crochemore et al. (2016) have shown that the simple linear scaling method provides results comparable to the more complex distribution mapping to correct the bias in precipitation ensemble forecasts".

**4. Page 13, line 16. General performance of watersheds 5 and 7 described is not clear**

Response : This issue was also raised by Reviewer 2 (his or her 3rd specific comment). This is a mistake. In fact, we clearly see that watershed 5 is not a good example in this sentence.
Page 16, lines 8-10 : The sentence "In general, corr-DSP outperforms ESP for the 1-month lead-time for watershed 5 and 7." has been replaced by "In general, corr-DSP outperforms ESP for the 1-month lead-time, with some

exceptions such as watershed number 5 in winter or watersheds number 3 and 9 during spring."

**6. Page 15, Figure 7. PIT histograms and not rank histograms, I expect.**

Response : Thank you for highlighting this mistake. Page 17, Figure 7 : The words "Rank histograms" have been replaced by "PIT histograms".

**7. Page 16, line 1-2. The problem of underdispersion of the bias-corrected ensemble could be elaborated. There is a general overestimation of precipitation cf. Fig. 2. In this case, linear scaling will produce a bias-corrected ensemble with smaller dispersion than the raw ensemble.**

Response : We agree. The linear scaling method modifies the dispersion of precipitation forecasts and can influence the dispersion of streamflow forecasts.
Page 18, line 23-24 : The sentence "As precipitation is most of the time overpredicted, as shown in Figure 2, bias corrected precipitation forecasts exhibit a lower dispersion than raw forecasts. This can explain the smaller dispersion of the volume forecasts." has been added.

**4    Additional changes in the manuscript**

The additional changes include all typo corrections, sentences rephrasing or additional information.

Page 1 , line 1 : Some complementary information have been added, now reads "Hydro-power production requires optimal dam and reservoir management to prevent flooding damage and avoid operation losses. "
Page 2 , line 14-16 : The sentence has been reformulated as follows "Examples include multiple regression type of models treated in a Bayesian perspective (e.g. Wang et al., 2009) or a frequentist framework (e.g. Moradkhani and Meier, 2010; Sveinsson et al., 2008)".
Page 2 , line 22-23 : The complementary information "either observed or simulated" has been added.
Page 3, line 7 : The word "Nordic" has been changed for "northern".
Page 3, line 30 : The word "Mutlimodel" has been corrected by "Multimodel".
Page 4, line 10 : The word "Predicition" has been corrected by "Prediction".
Page 4 , line 22-24 : The sentence has been added "Hydro-Québec is a government owned corporation that produces and distributes electricity in the Province of Québec. The installed hydroelectricity capacity of Hydro-Québec is more than 36 000MW." and the next sentence adapted, now reads " Together, the ten watersheds under study represent more than 8750 MW".
Page 6, Table 1 : Update of the values and footnote.
Page 9 , line 23 : Add-on "and optimization of electricity production".
Page 9, line 26-27 : Add-on "and reservoir operation".
Page 9 , line 33 to page 10, line 5 : Reformulation, now reads "The verification set should be as large as possible, in order to ensure statistical significance of the results. It should also be homogeneous. However, in reality, forecasts characteristics change depending on the period of the year and contradictory behaviors can balance each other out."
Page 10, line 11 : "tools" switched to "diagnostic devices".
Page 10 , line 12 : "score" switched to "scoring rule".
Page 10, line 27-28 : The sentence "PIT histograms are preferred over rank histograms herein because of the changing number of members (see section 3)." has been deleted.
Page 10 , line 31 : Typo correction, "of" switched to "for".
Page 11 , line 3 : Reformulation change, "presented in equation 2" switched to "expressed as".
Page 12 , Equations 3 and 4 : $k$ index has been added into the equations and the descriptive paragraph line 7.
Page 12 , line 8 : Correction "considered for the bias estimation" instead of "in the verification set".
Page 12, line 8-11 : Reformulation, now reads "We therefore assume an additive bias for monthly mean minimal and maximal temperature: it is the mean error. For monthly precipitation, bias is defined as the ratio of the forecasts mean to the mean observed accumulation. A multiplicative bias is then assumed for this meteorological variable."
Page 16 , line 23 : Terminology modification to be in line with technical terms used in section 4 for reliability diagram description. The modified sentence is "However, small differences between the observed coverage frequency and the theoretical confidence level remain."
Page 17 , Figure 6 : Modification of the axis labels to be in line with the definition of the reliability description given at Section 4.
Page 18 , line 25 : "boxplot" changed to plural form "boxplots".
Page 19, line 6 : Type, "analyse" switch to "analysis".

Page 20 , line 2 : Typo correction "CRPSS" instead of "CRPSSS".

[revised manuscript text omitted]